# Simultaneous estimation of bi-directional causal effects and heritable confounding from GWAS summary statistics

Liza Darrous [1,2,4], Ninon Mounier [1,2,4] & Zoltán Kutalik [1,2,3 ✉]

Mendelian Randomisation (MR) is an increasingly popular approach that estimates the causal effect of risk factors on complex human traits. While it has seen several extensions that relax its basic assumptions, most suffer from two major limitations; their under-exploitation of genome-wide markers, and sensitivity to the presence of a heritable confounder of the exposure-outcome relationship. To overcome these limitations, we propose a Latent Heritable Confounder MR (LHC-MR) method applicable to association summary statistics, which estimates bi-directional causal effects, direct heritabilities, and confounder effects while accounting for sample overlap. We demonstrate that LHC-MR outperforms several existing MR methods in a wide range of simulation settings and apply it to summary statistics of 13 complex traits. Besides several concordant results with other MR methods, LHC-MR unravels new mechanisms (how disease diagnosis might lead to improved lifestyle) and reveals new causal effects (e.g. HDL cholesterol being protective against high systolic blood pressure), hidden from standard MR methods due to a heritable confounder of opposite effect direction.

[1] University Center for Primary Care and Public Health, University of Lausanne, Lausanne, Switzerland. [2] Swiss Institute of Bioinformatics, Lausanne, Switzerland. [3] Department of Computational Biology, University of Lausanne, Lausanne, Switzerland. [4]These authors contributed equally: Liza Darrous, Ninon Mounier. ✉email: zoltan.kutalik@unil.ch

The identification of frequent risk factors and the quantification of their impact on common diseases is a principal quest for public health policy makers. Epidemiological studies aim to address this issue, but they are most often based on observational data due to their abundance over the years. Despite major methodological advances, a large majority of such studies have inherent limitations and suffer from confounding and reverse causation[1,2]. For these reasons, many of the reported associations found in classical epidemiological studies are mere correlates of disease risk, rather than causal factors directly involved in disease progression. Due to this, additional evidence is required before developing public health interventions in a bid to reduce the future burden of diseases. While well-designed and carefully conducted randomised control trials (RCTs) remain the gold standard for causal inference, they are exceedingly expensive, time-consuming, may not be feasible for ethical reasons, and have high failure rates[3,4].

Mendelian randomisation (MR), a natural genetic counterpart to RCTs, is an instrumental variable (IV) technique used to infer the strength of a causal relationship between a risk factor ($X$) and an outcome ($Y$)[5]. To do so, it uses genetic variants ($G$) as instruments and relies on three major assumptions (see Supplementary Fig. 1): (1) Relevance—$G$ is robustly associated with the exposure. (2) Exchangeability—$G$ is not associated with any confounder of the exposure-outcome relationship. (3) Exclusion restriction—$G$ is independent of the outcome conditional on the exposure and all confounders of the exposure-outcome relationship (i.e. the only path between the instrument and the outcome is via the exposure).

The advantage of the MR approach is that for most heritable exposures, dozens (if not hundreds) of genetic instruments are known to date thanks to well-powered genome-wide association studies (GWASs). Each instrument can provide a causal effect estimate, which can be combined with others, by using an inverse variance-weighting (IVW) scheme (e.g. Burgess et al.[6]). However, the last assumption is particularly problematic, because genetic variants tend to be pleiotropic, i.e. exert effect on multiple traits independently. Still, it can be shown that if the instrument strength is independent of the direct effect on the outcome (InSIDE assumption) and the direct effects are on average zero, IVW-based methods will still yield consistent estimates. Methods, such as MR-Egger[7], produce consistent estimates even if direct effects are allowed to have a non-zero offset. The third assumption can be further reduced to assuming that >50% of the instruments (or in terms of their weight) are valid (median-based estimators[8]) or that zero-pleiotropy instruments are the most frequent (mode-based estimators[9]).

The InSIDE assumption (i.e. horizontal pleiotropic effects ($G \rightarrow Y$) are independent of the direct effect ($G \rightarrow X$)) is reasonable if the pleiotropic path $G \rightarrow Y$ does not branch off to $X$. However, if there is such a branching off, the variable representing the split is a confounder of the $X - Y$ relationship and we fall back on the violation of the second assumption (exchangeability), making it the most problematic. Therefore, in this paper, we extend the standard MR model to incorporate the presence of a latent (i.e. unmeasured) heritable confounder ($U$) and estimate its contribution to traits $X$ and $Y$, while simultaneously estimating the bi-directional causal effect between the two traits. Standard MR methods are vulnerable to such heritable confounders, since any genetic marker directly associated with the confounder may be selected as an instrument for the exposure. However, such instruments will have a direct effect on the outcome that is correlated to their instrument strength, violating the InSIDE assumption and biasing the causal effect estimate.

In this paper, we first introduce the extended MR model and derive the likelihood function for the observed genome-wide summary statistics (for $X$ and $Y$). We then test and compare the method against conventional and more advanced (such as CAUSE[10] and MR-RAPS[11]) MR approaches through extensive simulation settings, including several violations of the model assumptions. Finally, the approach is applied to association summary statistics (based on the UK Biobank and meta-analysis studies) of 13 complex traits to re-assess all pairwise bi-directional causal relationships between them.

## Results

**Overview of the method.** We set up a structural equation model (SEM) (Fig. 1) and derived how its parameters are linked to genome-wide association summary statistics of two studied complex traits. We then maximised the resulting likelihood function in order to estimate bi-directional causal effects between them (for details see Methods), in addition to inferring direct heritabilities for $X$ and $Y$, confounder effects, cross-trait and individual trait LD-score intercepts and the polygenicity for $X$ and $Y$. All SNPs associated with the heritable confounder ($U$) are indirectly associated with $X$ and $Y$ with effects that are proportional (ratio $q_y/q_x$). SNPs that are directly associated with $X$ (and not with $U$) are also associated with $Y$ with proportional effects (ratio $1/\alpha_{x \rightarrow y}$). Finally, SNPs that are directly $Y$-associated are also $X$-associated with a proportionality ratio of $1/\alpha_{y \rightarrow x}$. These three groups of SNPs are illustrated on the $\beta_x$-vs-$\beta_y$ scatter plot (Supplementary Fig. 2). In simple terms, the aim of our method is to identify the different clusters, estimate the slopes and distinguish which corresponds to the causal- and confounder effects. In this paper, we focus on the properties of the maximum likelihood estimates (MLEs) (and their variances) for the bi-directional causal effects arising from our SEM.

**Simulation results.** We started off with a realistic simulation setting of 234,000 SNPs on chromosome 10 (LD patterns used from the UK10K panel) and 50,000 samples for both traits. Traits $X$, $Y$ and confounder $U$ had average polygenicity ($\pi_x = 5 \times 10^{-3}$, $\pi_y = 1 \times 10^{-2}$, $\pi_u = 5 \times 10^{-2}$), with substantial direct heritability for $X$ and $Y$ ($h_x^2 = 0.25$, $h_y^2 = 0.2$), mild confounding on $X$ and $Y$ ($t_x = 0.16$, $t_y = 0.11$, where $t_x = \sqrt{h_u^2 \cdot q_x^2}$ and $t_y = \sqrt{h_u^2 \cdot q_y^2}$), and a causal effect between $X$ and $Y$ ($\alpha_{x \rightarrow y} = 0.3$, $\alpha_{y \rightarrow x} = 0$). Note that with these settings, SNPs associated with $U$ would violate the InSIDE assumption but might still be used by conventional MR methods. Under this standard setting, there were no genome-

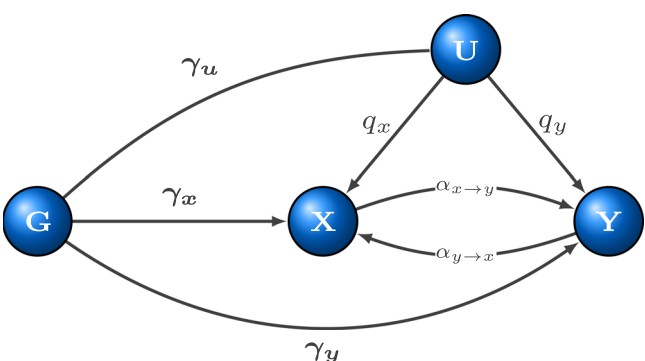

**Fig. 1 Schematic representation of the extended structural equation model (SEM).** $X$ and $Y$ are two complex traits under scrutiny with a latent (heritable) confounder $U$ with causal effects $q_x$ and $q_y$ on them. $G$ represents genetic variants, with effects $\gamma_x$, $\gamma_y$ and $\gamma_u$, respectively. Traits $X$ and $Y$ have causal effects on each other, which are denoted by $\alpha_{x \rightarrow y}$ and $\alpha_{y \rightarrow x}$.

wide significant SNPs for standard MR methods, and estimates derived using SNPs with a $p$-value $< 5 \times 10^{-6}$ showed a downward bias for all MR methods (Fig. 2a). MR-RAPS using filtered SNPs ($p$-value $< 5 \times 10^{-4}$) was similarly downward biased whereas MR-RAPS using the entire set of SNPs was upward biased with the least amount of variance compared to all methods including LHC-MR. LHC-MR in this scenario slightly overestimated the causal effect in comparison but had the smallest RMSE after MR-RAPS (0.13 vs 0.06, Supplementary Data 1).

We ran all our simulation scenarios with a smaller and a larger sample size (50,000 and 500,000) and observed that the relative performance of the methods were in some cases sample size specific. Smaller sample sizes often meant that standard MR methods had little to no IVs reaching genome-wide (GW) significance and hence we were forced to use IVs from less stringent thresholds ($< 5 \times 10^{-4}$ and $< 5 \times 10^{-6}$). Therefore, the causal effects were estimated with a substantial downward bias due to weak instrument bias (and winner's curse). LHC-MR in these cases was able to estimate the causal effect with less bias but with a larger variance compared to most standard MR methods—still outperforming them in terms of RMSE in most settings. In the larger sample size setting, standard MR methods had IVs for every threshold cutoff. However, a pattern also observed with smaller sample sizes—but to a lesser extent—emerged, where the causal estimates of some methods changed (either in mean or in variance, most noticeably observed in weighted median and IVW) as the threshold became more stringent. This is of particular concern and highlights that while in this simulation setting the $5 \times 10^{-8}$ threshold may have optimally cancelled out the different biases for IVW (downward bias due to winner's curse and weak instrument bias, upward bias due to genetic confounding), its estimate remains strongly setting-dependent. LHC-MR performed reasonably well, exhibiting lower RMSE than most other methods, except for IVW and MR-RAPS for the $5 \times 10^{-4}$ threshold (Supplementary Fig. 4a). However, we observed that the performance of MR-RAPs is particularly setting and threshold dependent.

Furthermore, unequal sample sizes for the two traits showed an underestimation of the causal effects for almost all MR methods, while LHC-MR remained the most accurate in the case where $n_x$ (50,000) was smaller than $n_y$ (500,000). However, the performances in the reverse scenario, where $n_x$ was larger in size, were akin to the large sample size standard setting, where only IVW and filtered MR-RAPS ($< 5 \times 10^{-4}$) showed superior performance to LHC-MR both in terms of bias and variance (see Supplementary Fig. 5).

When testing scenarios in the absence of a causal or a confounder effect (imitating the classical MR assumptions), with a smaller causal effect ($\alpha_{x \to y} = 0.1$), or with both forward- and reverse causal effects, we note that LHC-MR outperforms the standard MR methods as well as MR-RAPS in all these scenarios.

When there was no causal effect ($\alpha_{x \to y} = 0$), LHC-MR had the smallest bias out of all the methods in both sample sizes (0.004 in both, Supplementary Fig. 6a and Supplementary Fig. 7a). The variance of the LHC-MR estimates in the larger sample size was much lower (0.0001 vs 0.01), similarly the other methods had a smaller variance in the larger sample size and had more clearly seen upward biased estimates. The increased upward bias of standard MR methods is due to the fact that confounder-associated SNPs could only be detected in the larger sample size and those lead to positive bias (due to the concordant effect of the confounder on the two traits). Note that the variances of standard MR methods are low simply because, in these settings, we were forced to lower the instrument selection threshold, hence artificially included many (potentially invalid) instruments, which lowers the estimator variance while increasing bias. MR-RAPS

greatly overestimates the causal effects when the sample size is larger.

In the absence of a confounder effect, there is not much of a difference between the two sample sizes; standard MR methods have a large variance and are downward biased, LHC-MR is less biased compared to them but MR-RAPS performs best with the least bias and variance when all the SNPs are used as instruments (Supplementary Fig. 6b and Supplementary Fig. 7b). Trying a smaller causal effect led to an upward bias for all MR methods including both filterings of MR-RAPS in the larger sample size. Alternately, when $n_x = n_y = 50,000$, the MR methods are downward biased (Supplementary Figs. 6c and 7c). Lastly, when a (negative) reverse causal effect is introduced, all MR methods and MR-RAPS are negatively biased in their estimation of the causal effect (see Fig. 2b). LHC-MR has a much smaller bias for the forward causal effect estimate in this case, and a generally small bias for the reverse causal effect in both sample sizes (0.05 for $n = 50,000$ and 0.03 for $n = 500,000$, Supplementary Fig. 4b).

Increasing the indirect genetic effects, by intensifying the contribution of the confounder to $X$ and $Y$ ($t_x = 0.41, t_y = 0.27$), led to a general overestimation of the causal effects by all methods including LHC-MR, though more drastically seen in standard MR methods and MR-RAPS in the larger sample size, when there is sufficient power to pick up these confounder-associated SNPs. The causal effect estimates of standard MR methods in the smaller sample size were much less affected by the presence of a strong confounder compared to LHC-MR and MR-RAPS (Supplementary Fig. 8). The reason for this is that the confounder-associated SNPs remain undetectable at lower sample size and hence instruments will not violate the classical MR assumptions.

Further testing the effects of the confounder trait on the causal estimation, we tested the impact of confounders with opposite effects on $X$ and $Y$. We observe a major underestimation of the causal effects for standard MR methods as well as MR-RAPS, whereas LHC-MR performs better for both sample sizes (RMSE $= 0.01$ and $0.1$ for larger and smaller $n$ respectively), see Fig. 2c and Supplementary Fig. 4c.

Our LHC-MR method is influenced by the unlikely scenario of extreme polygenicity for traits $X$, $Y$ and $U$, and it suffers from increased bias and variance regardless of sample size (see Supplementary Fig. 9). Standard MR methods as well as filtered MR-RAPS underestimated the causal effect when $n = 50,000$. Some also underestimated $\alpha_{x \to y}$ when $n = 500,000$, with the exception of IVW, Mode and filtered MR-RAPS, that outperformed the rest. Decreasing the proportion of confounder-associated SNPs to 1% only, does not seem to affect our method and shows similar results to the standard setting (Supplementary Fig. 10).

Furthermore, we simulated summary statistics, where (contrary to our modelling assumptions) the $X - Y$ relationship has two confounders, $U_1$ and $U_2$. When the ratio of the causal effects of these two confounders on $X$ and $Y$ ($q_y^{(1)}/q_x^{(1)}$ and $q_y^{(2)}/q_x^{(2)}$, respectively) agreed in sign, the corresponding causal effects of standard MR methods were overestimated in the larger sample size and, conversely, underestimated in the smaller sample size (Supplementary Figs. 11a and 12a). LHC-MR and weighted median performed better however in the larger sample size and had a bias of 0.03 and 0.07, respectively. However, when the signs were opposite ($q_x^{(1)} = 0.3, q_y^{(1)} = 0.2$ for $U_1$ and $q_x^{(2)} = 0.3, q_y^{(2)} = -0.2$ for $U_2$), conventional MR methods and MR-RAPS in this case almost all underestimated the causal effect regardless of sample size. LHC-MR outperformed them both in the larger sample size (bias of 0.007) and in the smaller sample size (bias of $-0.003$), see Supplementary Figs. 11b and 12b.

Finally, we explored how sensitive our method is to different violations of our modelling assumptions. First, we simulated

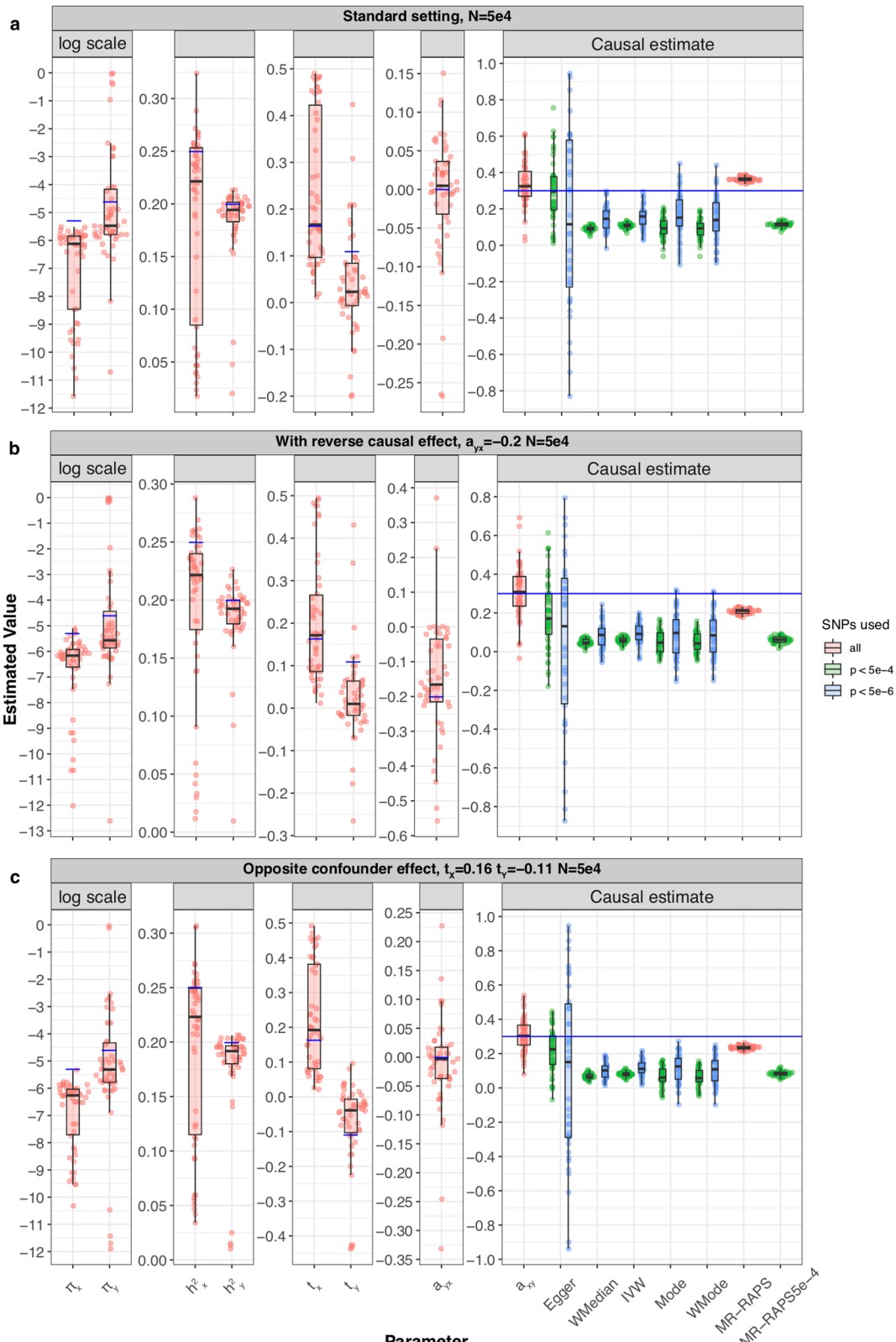

**Fig. 2 Simulation results under various scenarios.** These modified Sina-boxplots represent the distribution of parameter estimates from 50 different data generations under various conditions. For each generation, standard MR methods as well as our LHC-MR were used to estimate a causal effect. In the boxplots, the lower and upper hinges correspond to the first and third quartiles, the middle bar corresponds to the median, whereas the upper whisker is the largest dataset estimate smaller than 1.5×inter-quartile range above the third quartile. The lower whisker is defined analogously. The true values of the parameters used in the data generations are represented by the blue dots/lines. **a** Estimation under standard settings ($\pi_x = 5 \times 10^{-3}$, $\pi_y = 1 \times 10^{-2}$, $\pi_u = 5 \times 10^{-2}$, $h_x^2 = 0.25$, $h_y^2 = 0.2$, $h_u^2 = 0.3$, $t_x = 0.16$, $t_y = 0.11$). **b** Addition of a reverse causal effect $\alpha_{y \to x} = -0.2$. **c** Confounder with opposite causal effects on $X$ and $Y$ ($t_x = 0.16$, $t_y = -0.11$).

summary statistics when the underlying non-zero effects come from a non-Gaussian distribution. Interestingly, we observed that, for the smaller sample size, the variance of the causal effect estimate was dependent on the kurtosis for most MR methods. LHC-MR estimations yielded slightly more pronounced upward bias than IVW, while still exhibiting the lowest RMSE among all methods (Fig. 3a). Similar results are seen in the larger sample size with smaller variance for all methods under all degrees of kurtosis except for IVW, which showed a better performance than LHC-MR (Supplementary Fig. 13a). Second, we simulated effect sizes coming from a three-component Gaussian mixture distribution (null/small/large effects), instead of the classical spike-and-slab assumption of our model. The smaller sample size estimates mirror those of the standard setting with $n$ also equal to 50,000 (see Fig. 3b). However, in the larger sample size, LHC-MR overestimates the causal effect. This bias could be due to the merging of true effect estimates with confounder effect leading to an overestimation of $\alpha_{x \to y}$ (Supplementary Fig. 13b). MR-Egger, IVW and filtered MR-RAPS have the smallest RMSE in this case.

*Comparing CAUSE and LHC-MR.* When running CAUSE on data simulated using the LHC-MR model framework in order to estimate a causal effect ($\gamma$ in their notation), we investigated three different scenarios, each with multiple data generations: one where the underlying model has a shared factor/confounder with effect on both exposure and outcome only, another where the underlying model has a causal effect of 0.3 only, a third where the underlying model has both a causal effect and a shared factor. The data generated using the LHC-MR model was done under the standard settings ($\pi_x = 5 \times 10^{-3}$, $\pi_y = 1 \times 10^{-2}$, $\pi_u = 5 \times 10^{-2}$, $h_x^2 = 0.25$, $h_y^2 = 0.2$, $h_u^2 = 0.3$, $t_x = 0.16$, $t_y = 0.11$, $\alpha_{x \to y} = 0.3$, $\alpha_{y \to x} = 0$, $m = 234,000$, $n_x = n_y = 50,000$). For each setting, 50 different replications were investigated.

In the case of an underlying shared effect only, CAUSE preferred the sharing model 100% of the time, and thus there was no causal estimation, however it underestimated both $\eta$ and $q$. When there was an underlying causal effect only, CAUSE preferred the causal model only 4% of the times, where it slightly underestimated the causal effect ($\hat{\gamma} = 0.241$). Although the true values of $\eta$ and $q$ are null in this scenario, the sharing model returned estimates for these two parameters overestimating them both (probably driven by their priors), as seen in Supplementary Fig. 14. In the third case, and in the presence of both, CAUSE preferred the sharing model in 48 of the 50 simulations, yet it underestimated $\eta$ (corresponding to $t_y/t_x$ for our model) but overestimated $q$ ($t_x^2/(t_x^2 + h_x^2)$ in our model) (mean of 0.566 and 0.222, respectively, where the true values are 0.667 and 0.097) showing a similar estimation pattern to the second case. Interestingly, for the larger sample size, CAUSE selects the correct model 100% of the time, but still underestimates $\gamma$, as shown in Supplementary Fig. 15.

In the reverse situation, where data was generated using the CAUSE framework (with parameters $h_1 = h_2 = 0.25$, $m = 97,450$, $N1 = N2 = 50,000$) and LHC-MR was used to estimate the causal effect, we saw the following results (see Supplementary Fig. 16). First, when we generated data in the absence of causal effect ($\gamma = 0$, $\eta = \sqrt{0.05}$, $q = 0.1$), CAUSE does extremely well in estimating a null causal effect 100% of the time. Standard MR methods yield a slight overestimation of the (null) causal effect with varying degrees of variance, whereas LHC-MR shows both a greater variance and an upward bias—still leading to a causal effect compatible with zero. Second, in the absence of a confounder combined with non-zero causal effect ($\gamma = \sqrt{0.05} = 0.22$, $\eta = 0$, $q = 0$), CAUSE underestimates the causal effect ($\hat{\gamma} = 0.18$) compared to LHC-MR which overestimates the causal effect: the mean of the estimates was 0.38 (over the

50 runs). Finally, in the presence of both a confounder and a causal effect ($\gamma = \sqrt{0.05}$, $\eta = \sqrt{0.05}$, $q = 0.1$), CAUSE slightly underestimates the causal effect ($\hat{\gamma} = 0.20$), whereas LHC-MR overestimates the effects and shows estimates reaching the boundaries 11 out of 50 times (mean of the converged $\hat{\gamma} = 0.39$ over the 39 data simulations, see Supplementary Fig. 16c)— indicating that this setting of the CAUSE model is not compatible with the LHC-MR model framework. Interestingly, classical MR methods outperform CAUSE in this case. Note that in the interest of run time we used less SNPs (than usual) for parameter estimations. The analysis of the three separate scenarios was repeated for a larger sample size of 500,000 (Supplementary Fig. 17), with more favourable results for LHC-MR. In the absence of a causal effect, we had similar results to the smaller sample size, whereas in the absence of a shared effect, LHC-MR estimates the causal effect accurately with a mean of 0.22, CAUSE underestimates it and the rest of the MR methods are less biased. In the presence of both causal and shared factor, CAUSE recovers the causal effect. IVW, unlike the other MR methods and CAUSE, is more affected by the presence of the confounder, while LHC-MR exhibits upward bias with a mean estimate of 0.27.

**Application to association summary statistics of complex traits.** We applied our LHC-MR and other MR methods to estimate all pairwise causal effects between 13 complex traits (156 causal relationships in both directions). Our results are presented as a heatmap in Fig. 4 (and are detailed in Supplementary Data 2). Further, we calculated the alternate set of estimated parameters that naturally results from our model (for reference see Section The observed association summary statistics and Supplementary Methods 1.4). Among trait pairs for which the exposure had sufficient heritability (>2.5%), the alternate parameters of 102 trait pairs were within the possible ranges mentioned in methods (i.e. the confounder and the exposure are interchangeable). However, for all of these pairs, the alternative parameter optima lead to lower direct- than indirect heritability, which we deem unrealistic. Therefore, we report only the primary set of estimated optimal parameters in the main results and provide the alternative parameters in the Supplementary Data 3. The comparison of the results obtained by LHC-MR and standard MR methods is detailed below and more extensively in Supplementary Data 4–5. In summary, LHC-MR provided reliable causal effect estimates for 132 out of 156 exposure traits (i.e. those exposures had an estimated total heritability greater than 2.5%). These estimates were compared to five different MR methods. Seventy-four causal relationships were deemed significant by LHC-MR. Furthermore, for 117 out of those 132 comparable causal relationships, our LHC-MR causal effect estimates were concordant (not significantly different) with at least two out of five standard MR methods' estimates.

By simply comparing the significance status and the direction of the causal effects between the methods, we see that LHC-MR agrees in sign and significance (or the lack there of) with at least 3 MR methods 77 times. For 31 relationships, LHC-MR results lead to different conclusions than those of standard MR methods. For 28 of those, LHC-MR identified a causal effect missed by all standard MR methods. For the other three, we observed a disagreement in sign: LDL has a negative effect on BMI according to weighted mode and weighted median, whereas we show a positive effect, HDL and LDL show a negative bi-directional causal effect for weighted mode but a positive bi-directional effect with LHC-MR. Despite the conflicting evidence for the causal relationship of LDL on BMI, studies have shown that the relationship between them is non-linear[12], possibly explaining the discrepancy between the results.

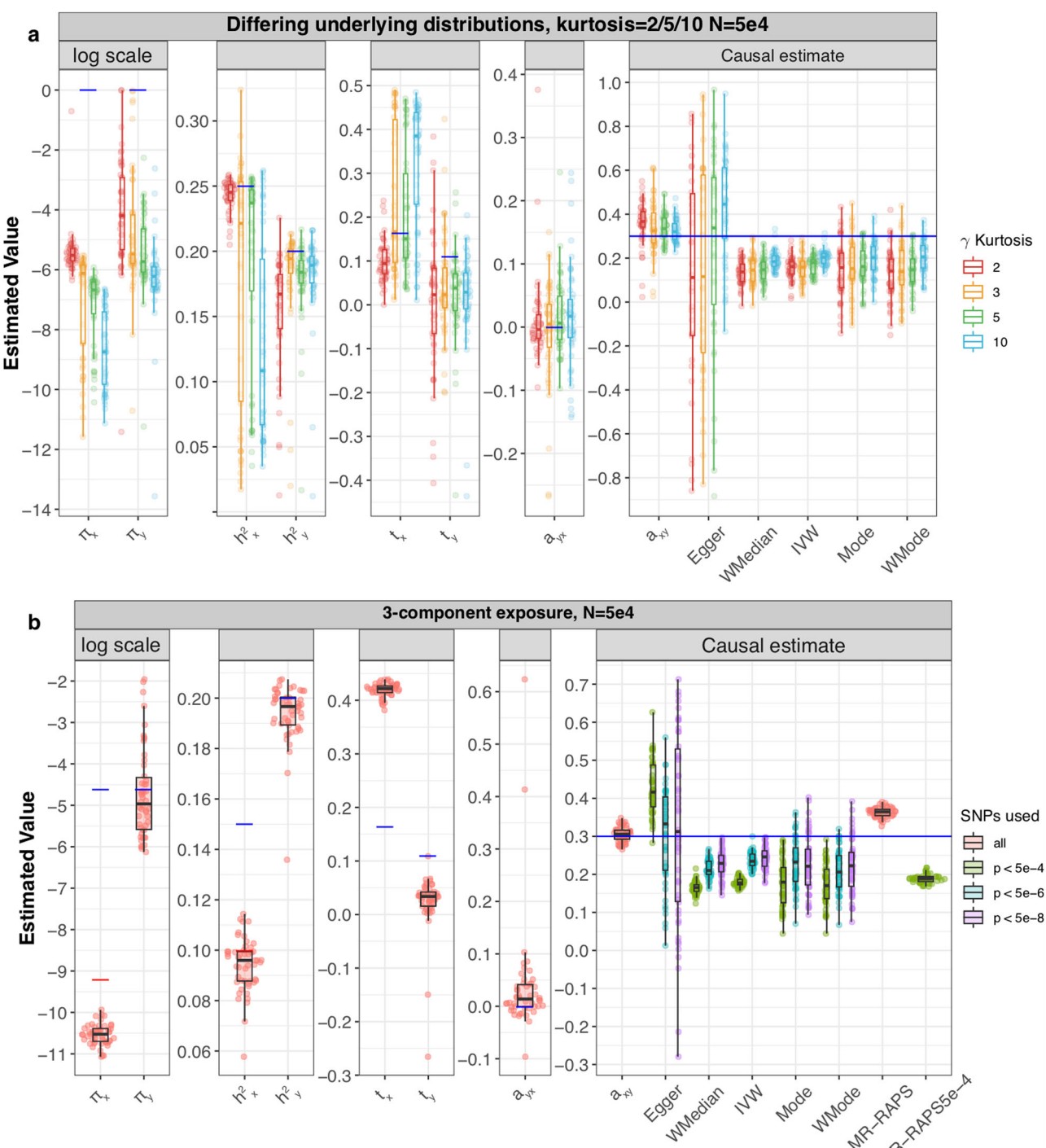

**Fig. 3 Simulation results under various scenarios.** These modified Sina-boxplots represent the distribution of parameter estimates from 50 different data generations under various conditions. For each generation, standard MR methods as well as our LHC-MR were used to estimate a causal effect. In the boxplots, the lower and upper hinges correspond to the first and third quartiles, the middle bar corresponds to the median, whereas the upper whisker is the largest dataset estimate smaller than 1.5×inter-quartile range above the third quartile. The lower whisker is defined analogously. The true values of the parameters used in the data generations are represented by the blue dots/lines. **a** The different coloured boxplots represent the underlying non-normal distribution used in the simulation of the three $\gamma_x, \gamma_y, \gamma_u$ vectors associated to their respective traits. The Pearson distributions had the same zero mean and skewness, however their kurtosis ranged between 2 and 10, including the kurtosis of 3, which corresponds to a normal distribution assumed by our model. The standard MR results reported had IVs selected with a p-value threshold of $5 \times 10^{-6}$. **b** Addition of a third component for exposure X, while decreasing the strength of U. True parameter values are in colour, blue and red for each component ($\pi_{x1} = 1 \times 10^{-4}, \pi_{x2} = 1 \times 10^{-2}, h_{x1}^2 = 0.15, h_{x2}^2 = 0.1$).

LHC-MR agreed with most MR estimates and confirmed many previous findings, such as increased BMI leading to elevated blood pressure[13,14], diabetes mellitus[15,16] (DM), myocardial infarction[17] (MI) and coronary artery disease[18] (CAD).

Furthermore, we confirmed previous results[19] that diabetes increases SBP ($\hat{\alpha}_{x \to y} = 0.39 - P = 1.70 \times 10^{-9}$).

Interestingly, it revealed that higher BMI increases smoking intensity, concordant with other studies[20,21]. It also showed the

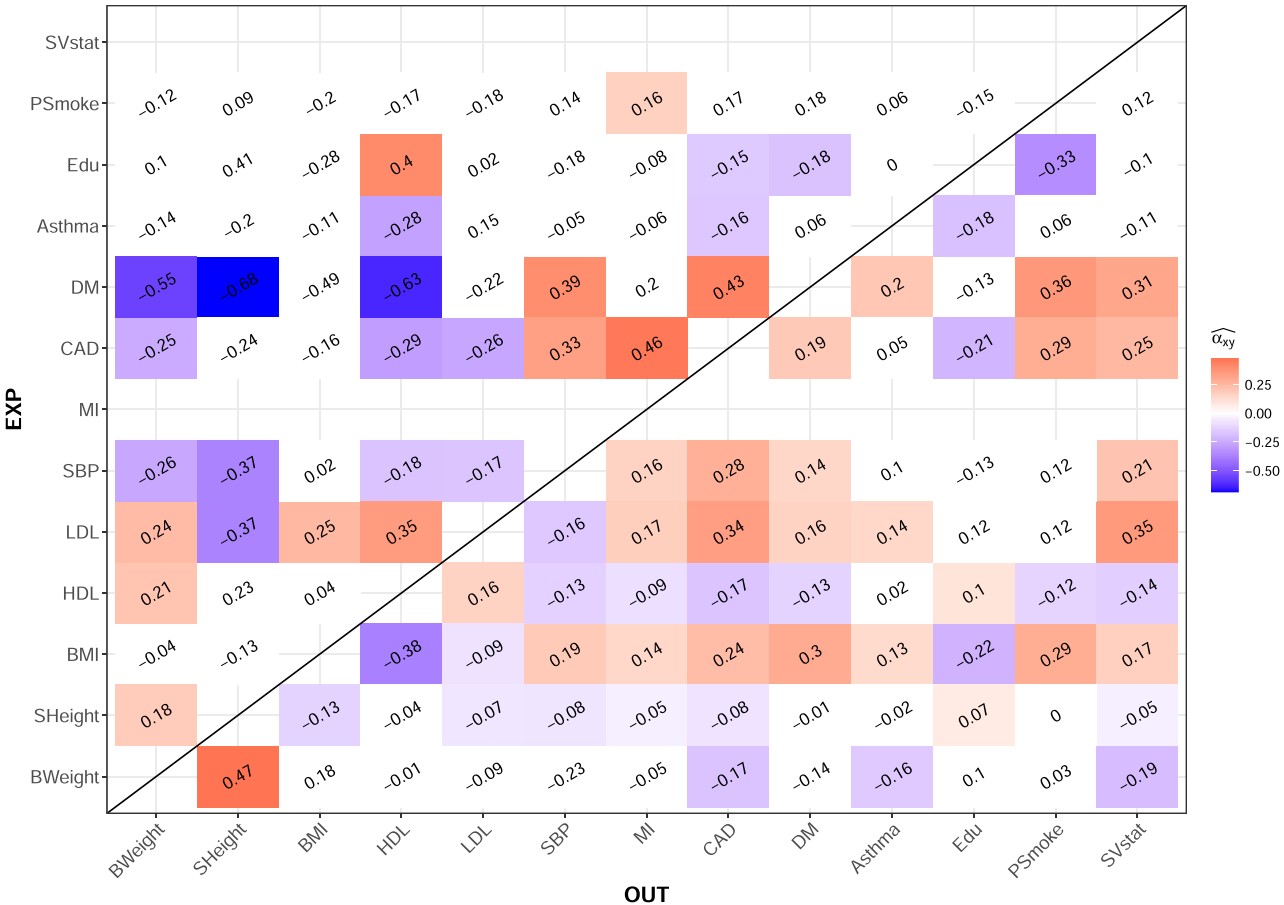

**Fig. 4 Heatmap representing the bi-directional causal relationship between the 13 UK Biobank traits.** The causal effect estimates in coloured tiles all have a significant *p*-value surviving Bonferroni multiple testing correction with a threshold of $3.2 \times 10^{-4}$. We did not report an estimated causal effect for exposures with an estimated total heritability less than 2.5%. White tiles show an absence of a significant causal effect estimate. BMI: Body Mass Index, BWeight: Birth Weight, CAD: Coronary Artery Disease, DM: Diabetes Mellitus, Edu: Years of Education, HDL: High-Density Lipoprotein, LDL: Low-Density Lipoprotein, MI: Myocardial Infarction, PSmoke: # of Cigarettes Previously Smoked, SBP: Systolic Blood Pressure, SHeight: Standing Height, SVstat: Medication-Simvastatin.

protective effect of education against a range of diseases (e.g. CAD and diabetes[22,23]) and risk factors such as smoking[24,25], in agreement with previous observational and MR studies. Probably reflecting lifestyle change recommendations by medical doctors upon disease diagnosis, statin use is greatly increased when being diagnosed with CAD, (systolic) hypertension, dislipidemia and diabetes as is shown by both LHC-MR and standard MR methods.

Furthermore, causal effects of height on CAD, DM and SBP have been previously examined in large MR studies[26,27]. LHC-MR, agreeing with these claims, did not find significant evidence to support the effect of height on DM, but did find a significant protective effect on CAD and SBP. However, unlike the first two, the relationship between height and SBP also revealed the existence of a confounder with causal effects $0.14 (P = 9.2 \times 10^{-11})$ and $0.11 (P = 3.39 \times 10^{-8})$ on height and SBP respectively. Another example of a trait pair for which LHC-MR found an opposite sign confounder effect is HDL and its protective effect on SBP. The confounder had a positive effect ratio of $t_y/t_x = 0.84$, opposing the negative causal effect of $\hat{\alpha}_{x \rightarrow y} = -0.13$ supported by observational studies[28]. This causal effect was not found by any other MR method.

It is important to note that while the effects of parental exposures on offspring outcomes can be seen as genetic confounding, LHC-MR would not be able to distinguish parental and offspring causal effects, because the LHC-MR model assumes

that there is no correlation between the genetic effects on the exposure and the genetic effects on the confounder (which is not the case for parental vs offspring traits). Thus, LHC-MR causal effect estimates are just as likely to reflect parental effects as any other MR method[29]. This may be the case, for example, for the detrimental effect of increased (parental) BMI on education (supported by longitudinal studies[30]), the positive effect of (parental) height on birth weight[31], or on education[32]. There are also some associations identified only by LHC-MR that might reflect parental effects: the negative causal effect of CAD on education or on birth weight, the positive impact of HDL on birth weight, or DM reducing height. All these pair associations uniquely found by LHC-MR are examples of LHC-MR's use of whole-genome SNPs instead of GW-significant SNPs only, as our estimates are of larger magnitude than those found by standard MR. Interestingly, for the CAD $\rightarrow$ birth weight relationship, LHC-MR revealed a confounder of opposite causal effects, which could have masked/mitigated the causal effect of standard MR methods.

A systematic comparison between IVW and LHC-MR has shown generally good agreement between the two methods, which is illustrated in Fig. 5. To identify discrepancies between our causal estimates and those of the standard MR results, we grouped the estimates into several categories, either non-significant *p*-value for both or either, significant with an agreeing sign for the causal estimate, or significant with a disagreeing sign. The diagonal (seen in Fig. 5) representing the agreement in

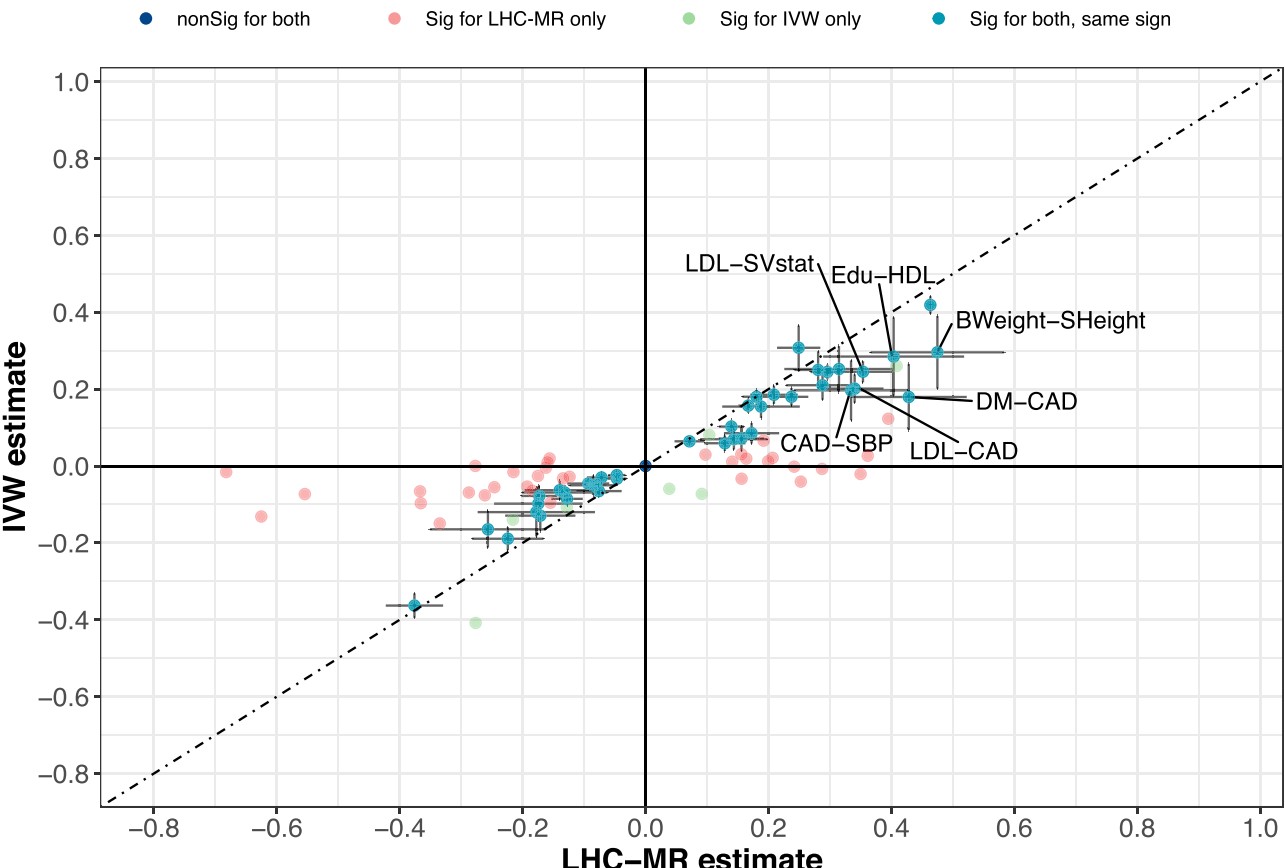

**Fig. 5 A scatter plot of the causal effect estimates between LHC-MR and IVW.** To improve visibility, non-significant estimates by both methods are placed at the origin, while significant causal estimates by both methods appear on the diagonal with 95% CI error bars. Pairs with an absolute value difference > 0.1 are labelled.

significance status and sign between the two methods, is heavily populated. On the other hand, 34 pairs have causal links that are significantly non-zero according to LHC-MR, but are non-significant for IVW, while the opposite is true for seven pairs. We believe that many of these seven pairs may be false positives, since four of them are picked up by no other MR method, two are confirmed by only one other method and the last one by two methods. Further comparisons of significance between LHC-MR estimates and the remaining standard MR methods can be found in Supplementary Table 2.

LHC-MR identified a confounder for 16 trait pairs out of the possible 78. In order to support these findings, we used EpiGraphDB[33,34] to systematically identify those potential confounders. EpiGraphDB could identify reliable confounders for ten out of the 16 trait pairs. Notably, for the birth weight–diabetes pair, the average epigraph confounder-effect ratio ($r_3/r_1$) clearly agreed in sign with our $t_y/t_x$ ratio, indicating that the characteristics of the confounder(s) evidenced by LHC-MR agree with those found in an exhaustive confounder search, and are mainly obesity-related traits (Supplementary Fig. 18a). Six other trait pairs showed mixed signs of different confounders, indicating the possibility of having heterogeneous confounders (Supplementary Fig. 18b-e). Finally, three trait pairs showed a disagreement between our estimated confounder-effect ratio and the bulk of those found by epighraphDB as seen in Supplementary Fig. 18f-j. However, at least one of the top ten potential confounders showed effects that are in agreement with our ratio for each of these pairs. Note that since the reported causal effects of the confounders on X and Y reported in EpiGraphDB are not

necessarily on the same scale, we do not expect the magnitudes to agree.

As described in the methods (Eq. (32)), genetic correlation can be computed from our estimated model parameters. To verify that the fitted LHC-MR model leads to a genetic correlation similar to the one obtained from LD-score regression[35] (LDSC), we compared whether the two approaches produce similar genetic correlation estimates. We did this by taking the estimated parameters obtained from the 200 block jackknife to estimate the genetic correlations between traits (and their standard errors), and plotted them against LD-score regression values as seen in Fig. 6. As expected, we observe an overall good agreement between the estimates of the two methods, with only six trait pairs differing in sign. Of these six, only 2 were nominally significantly different between the two methods (LDL → Asthma and LDL → DM). Further decomposition of the genetic covariance into heritable confounder-led or causal effect-led covariance revealed that most of the genetic covariance between traits can be attributed to bi-directional causal effects. A reason for this could be that confounders would need to have very strong effects to substantially contribute to the genetic correlation ($\approx t_x \cdot t_y$) compared to the bi-directional causal effects ($\approx \alpha_{x \to y}^2 \cdot h_x^2 + \alpha_{y \to x}^2 \cdot h_y^2$).

As for the comparison of LHC-MR against CAUSE for real trait pairs, we ran CAUSE on all 156 trait pairs (bi-directional), and extracted the parameter estimates that corresponded to the methods winning model. The p-value threshold was corrected for multiple testing and was equivalent to 0.05/156. Based on that threshold, the p-value that compared between the causal and the sharing model of CAUSE was used to choose one of the two. Then the parameters estimated from the winning model, $\gamma$ (only

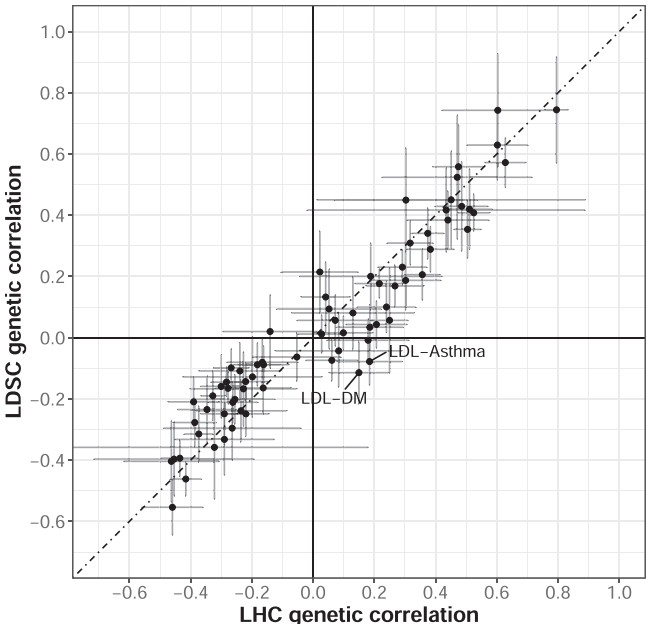

**Fig. 6 Scatter plot comparing the genetic correlation for each trait obtained from LDSC against the value calculated using parameter estimates from the LHC-MR model.** LHC-MR calculated genetic correlations from 200 parameter estimates generated during the block-jackknife procedure, where the mean values of these 200 estimates are shown here. A 95% CI for both method-calculation is shown for each point, and pairs with an absolute value difference > 0.2 are labelled. Values from both methods are reported in Supplementary Data 6.

for causal model), $\eta$ and $q$, were compared to their counterparts in LHC-MR. A visual comparison of LHC-MR's causal estimates and those of CAUSE can be seen in Supplementary Fig. 19.

Whenever the causal effect estimates were significant both for CAUSE and LHC-MR (30 causal relationships), they always agreed in sign (Supplementary Table 3) with a high Pearson correlation of 0.592. Calculating the correlation for their estimates regardless of significance yielded a smaller value of 0.377. When compared to the causal effect estimate from IVW, LHC-MR was strongly correlated (0.585), whereas CAUSE had a slightly weaker correlation (0.471) using all estimates.

Similarly, the significant confounder-effect ratio of LHC-MR ($t_y/t_x$) can be compared to the significant confounder-effect estimate of CAUSE ($\eta$) when a sharing model is chosen. These 12 confounding quantities by CAUSE and LHC-MR disagreed in sign for all but one trait pair (Height → MI), with a Pearson correlation compatible with zero ($-0.357$ (95% CI [$-0.77$, $0.27$])).

## Discussion
We have developed a structural equation (mixed-effects) model to account for a latent heritable confounder ($U$) of an exposure ($X$)–outcome ($Y$) relationship in order to estimate bi-directional causal effects between the two traits ($X$ and $Y$). The method, termed LHC-MR, fits this model to association summary statistics of genome-wide genetic markers to estimate various global characteristics of these traits, including bi-directional causal effects, confounder effects, direct heritabilities, polygenicities and population stratification.

We first demonstrated through simulations that in most scenarios, the method produces causal effect estimates with substantially less bias and variance (in the larger sample size) than other MR tools. The direction and magnitude of the bias of

classical MR approaches varied across scenarios and sample sizes. This bias was mainly influenced by two often opposite forces: downward bias resulting from winner's curse and weak instruments, and upward bias due to a positive confounder of the $X - Y$ relationship, evident in the larger sample size. In the scenario lacking a confounder (thus respecting all MR assumptions), MR methods were distinctly underestimating the causal effect, except for LHC-MR and to a better extent MR-RAPS. However, under standard settings with an added small heritable confounder and no reverse causality present, all classical MR methods still slightly underestimated the causal effect in the smaller sample size, except for the MR-RAPS estimate which was now overestimated. For the same standard setting scenario but in a larger sample size where confounder effects were more detectable, IVW had an estimation that was close to the true causal value chosen ($\alpha_{x \to y} = 0.3$) due to the opposite biases cancelling out. However, when the causal effect was set to be smaller ($\alpha_{x \to y} = 0.1$), the estimates of IVW became biased. More substantial violations of classical MR assumptions, such as the presence of negative-effect confounder or a negative reverse causal effect, led to more substantial biases that impacted all methods (including MR-RAPS) except LHC-MR.

Interestingly, in the smaller sample size, standard MR methods showed a slight decreasing trend in the variance of the causal effect estimate as the kurtosis of the underlying effect size distribution went up from 2 to 10. On the other hand, LHC-MR did not show a similar trend with growing kurtosis, and estimated the causal effect with a smaller bias. As confounder causal effects ($q_x$, $q_y$) increased, classical MR methods (except weighted ones) were prone to produce overestimated causal effects with at least twice the bias than that of LHC-MR, especially in the large sample size where the confounder-associated SNPs make it to the set of GW-significant instruments for all methods. Furthermore, mode-based estimators were robust to the presence of two concordant confounders, yet their bias was still 10-fold higher than LHC-MR's, and they did not perform as well in the presence of discordant confounders. In summary, LHC-MR was robust to a wide range of violations of the classical MR assumptions and was less impacted than standard MR methods. Thus it outperformed all MR methods in virtually all tested scenarios, many of which violated even its own modelling assumptions.

We then applied our method to summary statistics of 13 complex traits from large studies, including the UK Biobank. We observed a general trend in our results (in agreement with epidemiological studies) that higher BMI and LDL are risk factors for most diseases such as diabetes and CAD. We also note the protective effect HDL has on these same diseases. Moreover, we observe many disease traits increasing the intake of lipid-lowering medication (simvastatin), reflecting the recommendation/treatment of medical personnel following the diagnosis.

LHC-MR can have discordant results compared to other MR methods for many possible reasons. The positive causal effect of smoking on MI, diabetes on asthma, the protective impact of higher birth weight on asthma, or higher education on smoking intensity, all of which were missed by standard MR could reflect the increased power of LHC-MR with its use of full-genome SNPs as opposed to genome-wide significant SNPs of classical MR approaches.

Estimates from classical MR methods could also be impacted by sample overlap between the exposure and outcome datasets, whereas LHC-MR takes this into account. However, when using large sample sizes, the bias due to sample overlap is expected to be very small, and therefore not sufficient to explain any discrepancy in the results[36]. Another possible reason for the discrepancy between our findings and those of standard MR methods is the presence of a significant heritable confounder found by LHC-MR

with opposite effect to the estimated causal effect between the pair. These two opposite forces lead to association summary statistics that may be compatible with reduced (or even null) causal effect when the confounder is ignored. Possible examples of this scenario can be observed when (parental) traits, e.g. diabetes and CAD, act on birth weight. These pairs have a confounder of opposite effects, possibly related to (parental) obesity. Similarly, standard MR methods show little evidence for a causal effect of SBP on height, while our LHC-MR estimate is $-0.37$ ($P = 4.81 \times 10^{-8}$) which most probably reflects parental (maternal) effects as seen in previous studies[37,38]. The protective effect of HDL on SBP is another example where a confounder of opposite sign to that of the causal effect allows it to be uniquely found by LHC-MR. LHC-MR assumes no genetic correlation between the confounder and the direct effects on the exposure, which may be violated when the confounder is the same trait as the exposure, but in the parent. Such parental effects can mislead most MR methods[39], including ours, and hence we may observe biased results for traits such as BMI → education and HDL → birth weight.

Sixteen trait pairs showed a strong confounder effect, in the form of significant $t_x$ and $t_y$ estimates. These pairs were investigated for the presence of confounders using EpiGraphDB, and 10 of them returned possible confounders. The bulk of such pairs returned confounders with both agreeing and disagreeing effect directions on $X$ and $Y$, making it difficult to pinpoint a group of concordant and dominant confounders. However, for the birth weight-DM pair, where LHC-MR identifies a negative reverse causal effect and a confounder with effects $t_x = 0.10 (P = 6.77 \times 10^{-8})$ and $t_y = 0.15$ ($P = 3.13 \times 10^{-7}$) on birth weight and DM respectively, EpigraphDB confirmed several confounders related to body fat distribution and weight that matched in sign with our estimated confounder effect (Supplementary Fig. 18a). Note that EpiGraphDB causal estimates are not necessarily on the scale of SD outcome difference upon 1 SD exposure change scale, hence they are not directly comparable with the $t_y/t_x$ ratio, but are rather indicative of the sign of the causal effect ratio of the confounder. Furthermore, if EpigraphDB does not find a causal relationship between the trait pair in either directions, then it does not return any possible confounders of the two, a reason why only 10 out of 16 confounder-associated trait pairs returned any hits.

Lastly, our comparison of the genetic correlations calculated from our estimated parameters against those calculated from LD-score regression showed good concordance, confirming that the detailed genetic architecture proposed by our model is compatible with the observed genetic covariance. The major difference between the genetic correlation obtained by LD-score regression *vs* LHC-MR is that our model approximates all existing confounders by a single latent variable, which may be inaccurate when multiple ones exist with highly variable $t_y/t_x$ ratios. Furthermore, LHC-MR decomposed the observed genetic correlation into confounder and bi-directional causality driven components, revealing that most genetic correlations are primarily driven by bi-directional causal effects. Note that we have much higher statistical power to detect situations when the confounder effects are of opposite sign compared to the causal effects, because opposing genetic components are more distinct.

To our knowledge only two recent papers use similar models and genome-wide summary statistics. The LCV approach[40] is a special case of our model, where the causal effects are not included in the model, but they estimate the confounder effect mixed with the causal effect to estimate a quantity of genetic causality proportion (GCP). In agreement with others[10,41], we would not interpret non-zero GCP as evidence for causal effect. Moreover, in other simulation settings, LCV has shown very low power to detect causal effects (by rejecting GCP = 0) (Fig S15 in Howey et al.[42]). Another very recent approach, CAUSE[10],

proposes a structural equation mixed effect model similar to ours. However, there are several differences between LHC-MR and CAUSE: (a) we allow for bi-directional causal effects and model them simultaneously, while CAUSE is fitted twice for each direction of causal effect; (b) they first use an adaptive shrinkage method to integrate out the multivariable SNP effects and then go on to estimate other model parameters, while we fit all parameters at once; (c) CAUSE estimates the correlation parameter empirically; (d) we assume that direct effects come from a two-component Gaussian mixture, while they allow for larger number of components; (e) their likelihood function does not explicitly model the shift between univariate vs multivariate effects (i.e. the LD); (f) CAUSE adds a prior distribution for the causal/confounder effects and the proportion $\pi_u$, while LHC-MR does not; (g) to calculate the significance of the causal effect they estimate the difference in the expected log point-wise posterior density and its variance through importance sampling, whereas we use a simple block-jackknife method. Because of point (a), the CAUSE model can be viewed as a special case of ours when there is no reverse causal effect. We have the advantage of fitting all parameters simultaneously, while they only approximate this procedure. Although they allow for more than a two-component Gaussian mixture, for most traits with realistic sample sizes we do not have enough power to distinguish whether two or more components fit the data better. Therefore, we believe that a two-component Gaussian is a reasonable simplification. Due to the more complicated approach described in points (e-g), CAUSE is computationally more intense than LHC-MR, taking up to 1.25 CPU-hours in contrast to our 2.5 CPU-minute run time for a single starting point optimisation (which is massively parallelisable).

When we compared the performance of CAUSE and LHC-MR, we found that for large sample sizes both LHC-MR and CAUSE performed well not only when applied to data simulated by their own model, but also by the model of the other method. For smaller sample sizes, both methods performed poorly when applied to data generated by the other model. However, LHC-MR was less biased when applied to data generated by its own model than CAUSE was on data simulated based on its own model, where it provided rather conservative estimates. This is somewhat expected, since the primary aim of CAUSE is model selection and it is less geared towards parameter estimation, especially for settings where both sharing and causal effects are present (leading to very broad estimates). Also, CAUSE parameter estimates have shown to be somewhat sensitive to the choice of the prior.

Finally, when applying both LHC-MR and CAUSE to 156 complex trait pairs, we observed that the causal effects are reasonably well correlated (0.38 for all estimates, 0.59 for significant estimates) and agree in sign for trait pairs deemed significantly causal by either or both methods. In addition, LHC-MR causal estimates were more similar to those of IVW than the estimates provided by CAUSE. Surprisingly, when a confounding factor was identified by both methods, the confounder effects (LHC-MR $t_y/t_x$ ratio and CAUSE $\eta$ parameter) were uncorrelated. There are two possible explanations for this: (i) CAUSE may confuse/merge the confounder with the reverse causal effect, since it does not explicitly model the latter one. (ii) The two models assume different marginal effect size distributions, hence when multiple heterogeneous confounders exist, one method may detect one of the confounders, while the other method picks up the other confounder, depending on which has more similar genetic architecture to the assumed one.

Our approach has its own limitations, which we list below. Like any MR method, LHC-MR provides biased causal effect estimates if the input summary statistics are flawed (e.g. not corrected for complex population stratification, parental/dynasty effects). As

mentioned in the Methods section, our model is strictly-speaking unidentifiable and two distinct sets of parameters fit the data equally well, if the alternate set of parameters fall within the parameter ranges. As opposed to classical MR methods that give a single (biased) causal effect estimate, ours can detect and calculate the competing model. Due to biological considerations, from these competing models, we chose the one which yielded larger direct heritability than confounder-driven (indirect) heritability. Additional pointers to decide which parameter optimum we choose can be to pick the one with smaller magnitude of causal effects (large causal effects are unrealistic) or pick the one that includes causal effects that agree better with those of other MR methods.

LHC-MR is not an optimal solution for traits whose genetic architecture substantially deviates from a two-component Gaussian mixture of effect sizes. Also, for traits with low heritability (<2.5%), it is particularly important to compare the causal effect estimates to those from standard MR methods as results from LHC-MR may be less robust. In addition, trait pairs with multiple confounders with heterogeneous effect ratios can violate the single confounder assumption of the LHC model and can lead to biased causal effect estimates. Finally, LHC-MR, like other methods, is not immune to parental effects that are correlated with offspring effects. In such cases, the parental effect is grouped with the exposure (due to their strong genetic correlation) and not viewed as a confounder of the exposure-outcome relationship.

## Methods

**The underlying structural equation model.** Let $X$ and $Y$ denote continuous random variables representing two complex traits. Let us assume (for simplicity) that there is one heritable confounder $U$ of these traits. To simplify notation we assume that $E(X) = E(Y) = E(U) = 0$ and $Var(X) = Var(Y) = Var(U) = 1$. The genome-wide sequence data for $M$ sequence variants is denoted by $G = (G_1, G_2, ..., G_M)$. The aim of our work is to dissect the effects of the heritable confounding factor $U$ from the bi-directional causal effects of these two traits ($X$ and $Y$). For this we consider a model (see Fig. 1) defined by the following equations:

$$X = q_x \cdot U + \alpha_{y \to x} Y + G \cdot \gamma_x + e_x \qquad \text{with} \qquad e_x \sim \mathcal{N}(0, \nu_x^2) \qquad (1)$$

$$Y = q_y \cdot U + \alpha_{x \to y} X + G \cdot \gamma_y + e_y \qquad \text{with} \qquad e_y \sim \mathcal{N}(0, \nu_y^2) \qquad (2)$$

$$U = G \cdot \gamma_u + e_u \qquad \text{with} \qquad e_u \sim \mathcal{N}(0, \nu_u^2) \qquad (3)$$

where $\gamma_x, \gamma_y, \gamma_u \in \mathcal{R}^M$ denote the (true multivariable) direct effect of all $M$ genetic variants on $X$, $Y$ and $U$, respectively. All error terms ($e_x, e_y$ and $e_u$) are assumed to be independent of each other and normally distributed with variances $\nu_x^2, \nu_y^2$ and $\nu_u^2$, respectively.

Note that we do not include in the model reverse causal effects on the confounder ($X \to U$ and $Y \to U$). The reason for this is the following: Let $s_x$ and $s_y$ denote those causal effect of $X$ and $Y$ on $U$. We can see that by reparameterising the original model to $\alpha' := \alpha_{x \to y} + s_x \cdot q_y$, $\alpha' := \alpha_{y \to x} + s_y \cdot q_x$ and $q_x' := q_x/(1 - q_x \cdot s_x)$, $q_y' := q_y/(1 - q_y \cdot s_y)$, the genetic effects produced by the extended model with reverse causal effects on $U$ and the simpler model (Fig. 1) with the updated parameters are indistinguishable. Thus these extra parameters are not identifiable and the reparameterisation means that $\alpha_{x \to y}$ and $\alpha_{y \to x}$ in our model represent the total causal effects, some of which may be mediated by $U$.

Note that the model cannot be represented by classical directed acyclic graphs, as the bi-directional causal effects could form a cycle. However, the equations can be reorganised to avoid recursive formulation as follows:

$$X = q_x \cdot U + \alpha_{y \to x} \cdot \left( q_y \cdot U + \alpha_{x \to y} X + G \cdot \gamma_y + e_y \right) + G \cdot \gamma_x + e_x \qquad (4)$$

$$Y = q_y \cdot U + \alpha_{x \to y} \cdot \left( q_x \cdot U + \alpha_{y \to x} Y + G \cdot \gamma_x + e_x \right) + G \cdot \gamma_y + e_y \qquad (5)$$

$$U = G \cdot \gamma_u + e_u \qquad (6)$$

Regrouping the terms gives

$$(1 - \alpha_{y \to x}\alpha_{x \to y}) \cdot X = (q_x + \alpha_{y \to x} \cdot q_y) \cdot U + \alpha_{y \to x}(G \cdot \gamma_y) + G \cdot \gamma_x + (e_x + \alpha_{y \to x} \cdot e_y) \qquad (7)$$

$$(1 - \alpha_{x \to y}\alpha_{y \to x}) \cdot Y = (q_y + \alpha_{x \to y} \cdot q_x) \cdot U + \alpha_{x \to y}(G \cdot \gamma_x) + G \cdot \gamma_y + (e_y + \alpha_{x \to y} \cdot e_x) \qquad (8)$$

$$U = G \cdot \gamma_u + e_u \qquad (9)$$

Substituting $U$ into the first two equations yields

$$X = \frac{q_x + \alpha_{y \to x} \cdot q_y}{1 - \alpha_{y \to x}\alpha_{x \to y}} \cdot (G \cdot \gamma_u) + \frac{\alpha_{y \to x}}{1 - \alpha_{y \to x}\alpha_{x \to y}}(G \cdot \gamma_y) + \frac{1}{1 - \alpha_{y \to x}\alpha_{x \to y}}(G \cdot \gamma_x) + \epsilon_x \qquad (10)$$

$$Y = \frac{q_y + \alpha_{x \to y} \cdot q_x}{1 - \alpha_{x \to y}\alpha_{y \to x}} \cdot (G \cdot \gamma_u) + \frac{\alpha_{x \to y}}{1 - \alpha_{x \to y}\alpha_{y \to x}}(G \cdot \gamma_x) + \frac{1}{1 - \alpha_{x \to y}\alpha_{y \to x}}(G \cdot \gamma_y) + \epsilon_y \qquad (11)$$

with

$$\epsilon_x := \frac{e_x + \alpha_{y \to x} \cdot e_y + (q_x + \alpha_{y \to x} \cdot q_y) \cdot e_u}{1 - \alpha_{y \to x}\alpha_{x \to y}} \sim \mathcal{N}(0, i_x) \qquad (12)$$

$$\epsilon_y := \frac{e_y + \alpha_{x \to y} \cdot e_x + (q_y + \alpha_{x \to y} \cdot q_x) \cdot e_u}{1 - \alpha_{x \to y}\alpha_{y \to x}} \sim \mathcal{N}(0, i_y) \qquad (13)$$

where $i_x := (\nu_x^2 + \alpha_{y \to x}^2 \nu_y^2 + (q_x + \alpha_{y \to x} q_y)\nu_u^2)/(1 - \alpha_{y \to x}\alpha_{x \to y})^2$ and $i_y := (\nu_y^2 + \alpha_{x \to y}^2 \nu_x^2 + (q_y + \alpha_{x \to y} q_x)\nu_u^2)/(1 - \alpha_{x \to y}\alpha_{y \to x})^2$. Note that $i_x$ is equivalent to the LD-score regression intercept[43].

We model the genetic architecture of these direct effects with a spike-and-slab distribution, assuming that only $0 \le \pi_x, \pi_y, \pi_u \le 1$ proportion of the genome have a direct effect on $X$, $Y$, $U$, respectively, and these direct effects come from a Gaussian distribution. Namely,

$$\gamma_x = \zeta_x \odot \kappa_x \qquad \text{with} \qquad \kappa_x \sim \mathcal{N}(0, \sigma_x^2 \cdot I) \qquad \text{and} \qquad \zeta_x \sim \mathcal{B}_m(1, \pi_x) \qquad (14)$$

$$\gamma_y = \zeta_y \odot \kappa_y \qquad \text{with} \qquad \kappa_y \sim \mathcal{N}(0, \sigma_y^2 \cdot I) \qquad \text{and} \qquad \zeta_y \sim \mathcal{B}_m(1, \pi_y) \qquad (15)$$

$$\gamma_u = \zeta_u \odot \kappa_u \qquad \text{with} \qquad \kappa_u \sim \mathcal{N}(0, \sigma_u^2 \cdot I) \qquad \text{and} \qquad \zeta_u \sim \mathcal{B}_m(1, \pi_u) \qquad (16)$$

Here, $\odot$ denotes element-wise multiplication and $\mathcal{B}_m(1, q)$ the $m$ dimensional independent Bernoulli distribution. Further, we assume that all $\kappa_x, \kappa_y, \kappa_u$s are independent of each other and so are all $\zeta_x, \zeta_y, \zeta_u$s. We can refer to $h_x^2 := M \cdot \pi_x \cdot \sigma_x^2$ as the direct heritability of $X$, i.e. independent of the genetic basis of $U$ and $Y$. Similar notation is adapted for $U$ ($h_u^2 := M \cdot \pi_u \cdot \sigma_u^2$) and $Y$ ($h_y^2 := M \cdot \pi_y \cdot \sigma_y^2$). Note that when $q_x = 0$ and $q_y \ne 0$ (or vice versa), this means that there is no confounder $U$ present, but the genetic architecture of $Y$ (or $X$) can be better described by a three-component Gaussian mixture distribution.

We assume that the correlation (across markers) between the direct effects of a genetic variant on $X$, $Y$ and $U$ is zero, i.e. $cov(\gamma_x, \gamma_y) = cov(\gamma_x, \gamma_u) = cov(\gamma_u, \gamma_y) = 0$. Note that this assumption still allows for a potential correlation between the total effect of $G$ on $X$ and its horizontal pleiotropic effect on $Y$, but only due to the confounder $U$ and through the reverse causal effect $Y \to X$. As we argued above, this is a reasonable assumption, since the most plausible reason (apart from outcome-dependent sampling, which is out of the scope of this paper) for the violation of the InSIDE assumption may be one or more heritable confounder(s).

For simplicity, we also assume that the set of genetic variants with direct effects on each trait overlap only randomly, i.e. the fraction of the genome directly associated with both $X$ and $Y$ is $\pi_x \cdot \pi_y$, etc. This assumption is in line with recent observation that the bulk of observed pleiotropy can be explained by extreme polygenicity with random overlap between trait loci[44]. Note that uncorrelated effects (e.g. $cov(\gamma_x, \gamma_y) = 0$) do not ensure that the active variant sets overlap randomly, this is a slightly stronger assumption.

**The observed association summary statistics.** Let us now assume that we observe univariable association summary statistics for these two traits from two (potentially overlapping) finite samples $N_x$ and $N_y$ of size $n_x$, $n_y$, respectively. In the following, we will derive observed summary statistics in sample $N_x$ and then we will repeat the analogous exercise for sample $N_y$. Let the realisations of $X$, $Y$ and $U$ be denoted by $x$, $y$ and $u \in \mathcal{R}^{n_x}$. The genome-wide genetic data are represented by $G_x \in \mathcal{R}^{n_x \times M}$ and the genetic data for a single nucleotide polymorphism (SNP) $k$ tested for association is $g_k \in \mathcal{R}^{n_x}$. Note the distinction between the $k$-th column of $G_x$, which is the $k$-th sequence variant, in contrast to $g_k$, which is the $k$-th SNP tested for association in the GWAS. We assume that all SNP genotypes have been standardised to have zero mean and unit variance. The marginal effect size estimate for SNP $k$ of trait $X$ can then be written as $\widehat{\beta}_k^x = g_k' \cdot x/n_x$, which is a special case of univariable standard normal linear regression when both the outcome and the predictor is standardised to have zero mean and unit variance[43]. Note that $x'$ denotes the transpose of the column vector $x$. This can be further transformed as

$$\begin{aligned} \widehat{\beta}_k^x &= g_k' \cdot x/n_x \\ &= \frac{q_x + \alpha_{y \to x} \cdot q_y}{1 - \alpha_{y \to x}\alpha_{x \to y}} \cdot g_k' \cdot G_x \cdot \gamma_u/n_x + \frac{\alpha_{y \to x}}{1 - \alpha_{y \to x}\alpha_{x \to y}} \cdot g_k' \cdot G_x \cdot \gamma_y/n_x \\ &\quad + \frac{1}{1 - \alpha_{y \to x}\alpha_{x \to y}} \cdot g_k' \cdot G_x \cdot \gamma_x/n_x + g_k' \cdot \epsilon_x/n_x \end{aligned} \qquad (17)$$

By denoting the linkage disequilibrium (LD) between variant $k$ and all markers in

the genome with $\boldsymbol{\rho}_k = G'_x \cdot \boldsymbol{g}_k / n_x$ we get

$$\widehat{\beta}^x_k = \frac{q_x + \alpha_{y \to x} \cdot q_y}{1 - \alpha_{y \to x} \alpha_{x \to y}} \cdot \boldsymbol{\rho}'_k \cdot \boldsymbol{\gamma}_u + \frac{\alpha_{y \to x}}{1 - \alpha_{y \to x} \alpha_{x \to y}} \cdot \boldsymbol{\rho}'_k \cdot \boldsymbol{\gamma}_y + \frac{1}{1 - \alpha_{y \to x} \alpha_{x \to y}} \cdot \boldsymbol{\rho}'_k \cdot \boldsymbol{\gamma}_x + \eta^x_k$$

(18)

with $\eta^x_k := \boldsymbol{g}'_k \cdot \boldsymbol{\epsilon}_x / n_x \sim \mathcal{N}(0, i_x / n_x)$. Given the above-defined genetic effect size distribution the equation becomes

$$\widehat{\beta}^x_k = \frac{q_x + \alpha_{y \to x} \cdot q_y}{1 - \alpha_{y \to x} \alpha_{x \to y}} \cdot \underbrace{\boldsymbol{\rho}'_k \cdot (\boldsymbol{\zeta}_u \odot \boldsymbol{\kappa}_u)}_{z^{(u)}_k} + \frac{\alpha_{y \to x}}{1 - \alpha_{y \to x} \alpha_{x \to y}} \cdot \underbrace{\boldsymbol{\rho}'_k \cdot (\boldsymbol{\zeta}_y \odot \boldsymbol{\kappa}_y)}_{z^{(y)}_k}$$

$$+ \frac{1}{1 - \alpha_{y \to x} \alpha_{x \to y}} \cdot \underbrace{\boldsymbol{\rho}'_k \cdot (\boldsymbol{\zeta}_x \odot \boldsymbol{\kappa}_x)}_{z^{(x)}_k} + \eta^x_k$$

(19)

$$= \frac{q_x + \alpha_{y \to x} \cdot q_y}{1 - \alpha_{y \to x} \alpha_{x \to y}} \cdot z^{(u)}_k + \frac{\alpha_{y \to x}}{1 - \alpha_{y \to x} \alpha_{x \to y}} \cdot z^{(y)}_k + \frac{1}{1 - \alpha_{y \to x} \alpha_{x \to y}} z^{(x)}_k + \eta^x_k$$

Similarly, assuming that the LD structures ($\boldsymbol{\rho}_k$) in the two samples are comparable, for $\widehat{\beta}^y_k$ estimated in the other sample ($N_y$) we obtain

$$\widehat{\beta}^y_k = \frac{\alpha_{x \to y} \cdot q_x + q_y}{1 - \alpha_{x \to y} \alpha_{y \to x}} \cdot z^{(u)}_k + \frac{\alpha_{x \to y}}{1 - \alpha_{x \to y} \alpha_{y \to x}} \cdot z^{(x)}_k + \frac{1}{1 - \alpha_{x \to y} \alpha_{y \to x}} z^{(y)}_k + \eta^y_k \quad (20)$$

with $\eta^y_k \sim \mathcal{N}(0, i_y / n_y)$.

Therefore, the joint effect size estimates can be written as

$$\begin{pmatrix} \widehat{\beta}^x_k \\ \widehat{\beta}^y_k \end{pmatrix} = \frac{1}{1 - \alpha_{x \to y} \alpha_{y \to x}} \left( \begin{pmatrix} (\alpha_{y \to x} \cdot q_y + q_x) \\ (\alpha_{x \to y} \cdot q_x + q_y) \end{pmatrix} z^{(u)}_k + \begin{pmatrix} 1 \\ \alpha_{x \to y} \end{pmatrix} z^{(x)}_k + \begin{pmatrix} \alpha_{y \to x} \\ 1 \end{pmatrix} z^{(y)}_k \right) + \begin{pmatrix} \eta^x_k \\ \eta^y_k \end{pmatrix}$$

(21)

Following the same rational as the cross-trait LD-score regression[45], the noise term distribution is readily obtained

$$\begin{pmatrix} \eta^x_k \\ \eta^y_k \end{pmatrix} \sim \mathcal{N} \left( \begin{pmatrix} 0 \\ 0 \end{pmatrix}, \begin{pmatrix} i_x / n_x & \frac{n_{x \cap y}}{n_x \cdot n_y} \cdot r_{x,y} \\ \frac{n_{x \cap y}}{n_x \cdot n_y} \cdot r_{x,y} & i_y / n_y \end{pmatrix} \right)$$

(22)

where $r_{x,y}$ is the observational correlation between variables $X$ and $Y$ and $n_{x \cap y}$ is the size of the overlapping samples for $X$ and $Y$. Since both $n_{x \cap y}$ and $r_{x,y}$ cannot be estimated, we simply denote $i_{x,y} := r_{x,y} \cdot \frac{n_{x \cap y}}{\sqrt{n_x \cdot n_y}}$ as the only estimated parameter and parameterise the covariance term as $\frac{i_{x,y}}{\sqrt{n_x \cdot n_y}}$. Note that $i_{x,y}$ is the cross-trait LD-score regression intercept.

While the bivariate probability density function (PDF) of these summary statistics cannot be obtained analytically, we could derive its characteristic function (see Supplementary Methods 1.1), which is the product of some transformed version of the characteristic functions of $z^{(x)}_k, z^{(u)}_k, z^{(y)}_k$ and $(\eta^x_k, \eta^y_k)$, yielding

$$\varphi_{\left( \widehat{\beta}^x_k, \widehat{\beta}^y_k \right)} (v, w) = E\left[ \exp\left( i \cdot \left( v \cdot \widehat{\beta}^x_k + w \cdot \widehat{\beta}^y_k \right) \right) \right]$$

$$= \varphi_{z^{(u)}_k} \left( \frac{v \cdot (\alpha_{y \to x} \cdot q_y + q_x) + w \cdot (\alpha_{x \to y} \cdot q_x + q_y)}{1 - \alpha_{x \to y} \alpha_{y \to x}} \right)$$

$$\times \varphi_{z^{(x)}_k} \left( \frac{v + \alpha_{x \to y} \cdot w}{1 - \alpha_{x \to y} \alpha_{y \to x}} \right) \cdot \varphi_{z^{(y)}_k} \left( \frac{w + \alpha_{y \to x} \cdot v}{1 - \alpha_{x \to y} \alpha_{y \to x}} \right) \cdot \varphi_{(\eta^x_k, \eta^y_k)} (v, w)$$

(23)

Approximating the local correlations of SNP $k$ ($\boldsymbol{\rho}_k$) by a spike and slab distribution, parameterised by the fraction of non-zero correlations ($\pi_k$) and the variance of the non-zero correlations ($\sigma^2_k$), allows the derivation of a closed form expressions of $\varphi_{z^{(u)}_k}, \varphi_{z^{(x)}_k}$ and $\varphi_{z^{(y)}_k}$.

**Derivation of the likelihood function.** Given that the characteristic function can be analytically derived, we used the inversion theorem (for characteristic functions) to obtain the joint distribution of $\left( \widehat{\beta}^x_k, \widehat{\beta}^y_k \right)$ as

$$f_{\left( \widehat{\beta}^x_k, \widehat{\beta}^y_k \right)} (x, y) = \left( \frac{1}{2\pi} \right)^2 \cdot \int_{-\infty}^{\infty} \int_{-\infty}^{\infty} \exp(-i \cdot (x \cdot v + y \cdot w)) \cdot \varphi_{\left( \widehat{\beta}^x_k, \widehat{\beta}^y_k \right)} (v, w) \, dv \, dw$$

(24)

This integral can be efficiently computed by the Fast Fourier Transformation (FFT, see ref. [46] and references within. Detailed derivation is found in Supplementary Methods 1.2). To speed up computation, we bin SNPs according to their $\pi_k$ and $\sigma_k$ values which characterise the local LD distribution for each SNP $k$ ($10 \times 10$ bins with equidistant centres - see Supplementary Methods 1.3) and for SNPs in the same bin the PDF function is evaluated over a fine grid ($2^7 \times 2^7$ combinations) using the FFT.

To reduce the number of parameters we define $t_x := \sigma_u \cdot q_x$ and $t_y := \sigma_u \cdot q_y$ since $\sigma_u$ and $q_x$ are separately not identifiable, but only their product is. Extensive

simulations have shown that $\pi_u$ is unidentifiable, and hence is set to an arbitrary value of 0.1. For improved interpretability, we slightly reparameterise the likelihood function by using $h^2_x := \pi_x \cdot M \cdot \sigma^2_x, h^2_y := \pi_y \cdot M \cdot \sigma^2_y$. Since different SNPs are correlated we have to estimate the over-counting of each SNP. We choose the same strategy as LD-score regression[43] and weigh each SNP by the inverse of its restricted LD score, i.e. $w_k = 1 / \sum_{j=1}^{m_0} r^2_{jk}$, where $r_{jk}$ is the correlation between GWAS SNPs $k$ and $j$. The log-likelihood function is, thus, of the form

$$\log \left( \mathcal{L} \left( \boldsymbol{\theta} \middle| \begin{pmatrix} \widehat{\boldsymbol{\beta}}^x \\ \widehat{\boldsymbol{\beta}}^y \end{pmatrix} \right) \right) \propto \sum_{k=1}^{K} w_k \cdot f_k \left( \widehat{\beta}^x_k, \widehat{\beta}^y_k \right)$$

(25)

where $f_k \left( \widehat{\beta}^x_k, \widehat{\beta}^y_k \right)$ is the log-likelihood function value for SNP $k$. Parameters $\{n_x, n_y, m, \sigma_{k=1,\ldots,K}, \pi_{k=1,\ldots,K}\}$ are known and the other 11 parameters

$$\boldsymbol{\theta} = \{\pi_x, \pi_y, h^2_x, h^2_y, t_x, t_y, \alpha_{x \to y}, \alpha_{y \to x}, i_x, i_y, i_{x,y}\}$$

are to be estimated from the observed association summary statistics. In order to further speed up computation, we estimate the 11 parameters in two separate steps: we first estimate for each trait the parameters $\pi_x, i_x$ and $\pi_y, i_y$ (SNP polygenicity and LD-score intercept) and the total heritability (unlike the direct heritability obtained by the full-model of LHC-MR) by using a simplified model with only the trait of interest, without a second trait or confounder, e.g. we fit only $\pi_x, h^2_x$ and $i_x$ using $\widehat{\boldsymbol{\beta}}^x$ and assume that $\pi_x$ and $i_x$ do not change when two traits are taken into account. Note that $\pi_x$ may change slightly (decreasing from the total to direct polygenicity), but its value has little impact on the likelihood function. The estimates from the first step are then fixed for the parameter estimation of trait pairs in the second step. Since only $\pi_x, i_x$ and $\pi_y, i_y$ is fixed, the remaining parameters to estimate are now:

$$\boldsymbol{\theta} = \{h^2_x, h^2_y, t_x, t_y, \alpha_{x \to y}, \alpha_{y \to x}, i_{x,y}\}$$

It is key to note that our approach does not aim to estimate individual (direct or indirect) SNP effects, as these are handled as random effects. By replacing $U$ with $-U$ we swap the signs of both $t_x$ and $t_y$, therefore these parameters are unique only if the sign of one of them is fixed. Thus, we will have the following restrictions on the parameter ranges: $h^2_x, h^2_y, t_x$ are in $[0, 1]$, $t_y, \alpha_{x \to y}, \alpha_{y \to x}, i_{x,y}$ are in $[-1, 1]$.

**Likelihood maximisation and standard error calculation.** Our method, termed Latent Heritable Confounder Mendelian Randomisation (LHC-MR), maximises this likelihood function to obtain the MLE. Due to the complexity of the likelihood surface, we initialise the maximisation using 50 different starting points, where they come from a uniform distribution within the parameter-specific ranges mentioned above. We then choose parameter estimates corresponding to the highest likelihood of the 50 runs. Run time depends on the number of iterations during the maximisation procedure, and is linear with respect to the number of SNPs. It takes ~0.25 CPU-minute to fit the complete model to 50,000 SNPs with a single starting point.

Given the particular nature of the underlying directed graph, two different sets of parameters lead to an identical fit of the data, resulting in two global optima. The reason for this is the difficulty in distinguishing the ratio of the confounder effects ($t_y / t_x$) from the causal effect ($\alpha_{x \to y}$), as illustrated in Supplementary Fig. 2 by the slopes belonging to different SNP-clusters. More rigorously, it can be show that if $\{h_x, h_y, \alpha_{x \to y}, \alpha_{y \to x}, t_x, t_y\}$ is an optimum, then so will be $\{h'_x, h'_y, \alpha'_{x \to y}, \alpha'_{y \to x}, t'_x, t'_y\}$, where

$$h'_x = t_x + t_y \cdot \alpha_{y \to x}$$

(26)

$$h'_y = h_y$$

(27)

$$\alpha' = \frac{\alpha_{x \to y} + w}{1 + \alpha_{y \to x} \cdot w}$$

(28)

$$\alpha' = \alpha_{y \to x}$$

(29)

$$t'_x = h_x \cdot (1 + \alpha_{y \to x} \cdot w)$$

(30)

$$t'_y = -h_x \cdot w$$

(31)

with $w = t_y / t_x$ (for further derivations, see Supplementary Methods 1.4). This allows us to directly obtain both optima, even if the optimisation only revealed one of them. It happens very often that one of these parameter sets are outside of the allowed ranges and hence can be automatically excluded. If not, we keep track of both parameter estimates maximising the likelihood function. Note that, we call the one for which the direct heritability is larger than the indirect one, i.e. $h^2_x > t^2_x$, the primary solution. We show that for real data application this solution is far more plausible than the alternative optimum. Finally, note that such bimodality can be observed at different levels: (i) For one given data generation, using multiple starting points leads to different optima; (ii) LHC-MR applied to multiple different data generations for a fixed parameter setting can yield different optima. Both of

these situations are signs of the same underlying phenomenon and most often co-occur.

We implemented the block-jackknife procedure that is also used by LD-score regression to calculate the standard errors. For this we split the genome into 200 jackknife blocks and compute MLE in a leave-one-block-out fashion yielding $\widehat{\theta}^{(-i)}, i = 1, \ldots, 200$ estimates. The variance of the full SNP MLE is then defined as

$$Var(\widehat{\theta}) := \frac{m - m \cdot (1/200)}{m \cdot (1/200)} \cdot \frac{1}{200-1} \sum_{i=1}^{200} (\widehat{\theta}^{(-i)} - \overline{\widehat{\theta}})^2 = \sum_{i=1}^{200} (\widehat{\theta}^{(-i)} - \overline{\widehat{\theta}})^2 .$$

**Decomposition of genetic correlation**. Given the starting equations for $X$ and $Y$ (Eqs. (2)–(3)) we can calculate their genetic correlation as the ratio between their genetic covariance and variance (calculated from their heritabilities) as such:

$$corr(\delta_x, \delta_y) = \frac{(t_x + \alpha_{y \to x} t_y)(t_y + \alpha_{x \to y} t_x) + \alpha_{y \to x} h_y^2 + \alpha_{x \to y} h_x^2}{\sqrt{\left((t_x + \alpha_{y \to x} t_y)^2 + \alpha_{y \to x}^2 h_y^2 + h_x^2\right)\left((t_y + \alpha_{x \to y} t_x)^2 + \alpha_{x \to y}^2 h_x^2 + h_y^2\right)}}$$

(32)

The full details of the derivation is found in Supplementary Methods 1.5. Using our estimated parameters, we first calculate the correlation based on Eq. (32) and then compare them to those obtained by LD-score regression.

**Simulation settings**. First, we tested LHC-MR using realistic parameter settings with a mild violation of the classical MR assumptions. These standard parameter settings consisted of simulating $m = 234,000$ SNPs for two non-overlapping cohorts of equal size (for simplicity) of $n_x = n_y = 50,000$ for each trait. $X$, $Y$ and $U$ were simulated with moderate polygenicity ($\pi_x = 5 \times 10^{-3}$, $\pi_y = 1 \times 10^{-2}$, $\pi_u = 5 \times 10^{-2}$), and considerable direct heritability ($h_x^2 = 0.25$, $h_y^2 = 0.2$, $h_u^2 = 0.3$). $U$ had a confounding effect on the two traits as such, $q_x = 0.3$, $q_y = 0.2$ (resulting in $t_x = 0.16$, $t_y = 0.11$), and $X$ had a direct causal effect on $Y$ ($\alpha_{x \to y} = 0.3$), while the reverse causal effect from $Y$ to $X$ was set to null. Note that in this setting the total heritability of each of these traits is principally driven by direct effects and less than 10% of the total heritability is through a confounder and in case of $Y$ less than an additional 8% of its total heritability is through $X$. It is important to note that for each tested parameter setting, we generated 50 different datasets, and each data generation underwent a likelihood maximisation of Eq. (25) using 50 starting points, and produced estimated parameters corresponding to the highest likelihood (simplified schema in Supplementary Fig. 3).

In the following simulations, we changed various parameters of these standard settings to test the robustness of the method. We explored how increased sample size ($n_x = n_y = 500,000$) or differences in sample sizes (($(n_x, n_y) = (50,000, 500,000)$) and ($(n_x, n_y) = (500,000, 50,000)$)) influence causal effect estimates of LHC-MR and other MR methods. We also simulated data with no causal effect (or with no confounder) and then examined how LHC-MR estimates those parameters. Next, we varied our causal effects between the two traits by lowering $\alpha_{x \to y}$ to 0.1, and in another setting by introducing a reverse causal effect ($\alpha_{y \to x} = -0.1$). In addition, we tried to create extremely unfavourable conditions for all MR analyses by varying the confounding effects. We did this in several ways: (i) increasing $q_x$ and $q_y$ ($q_x = 0.75$, $q_y = 0.50$), (ii) having a confounder with causal effects of opposite signs on $X$ and $Y$ ($q_x = 0.3$, $q_y = -0.2$). We also drastically increased the proportion of SNPs with non-zero effect on traits $X$, $Y$ and $U$ ($\pi_x$, $\pi_y$ and $\pi_u = 0.1, 0.15, 0.2$ respectively). We also simulated data whereby the confounder has lower ($\pi_u = 0.01$) polygenicity than the two focal traits.

Finally, we explored various violations of the assumptions of our model (see Methods Section). First, we introduced two confounders in the simulated data, once with causal effects on $X$ and $Y$ that were concordant ($t_x^{(1)} = 0.16, t_y^{(1)} = 0.11, t_x^{(2)} = 0.22, t_y^{(2)} = 0.16$) in sign, and another with discordant effects ($t_x^{(1)} = 0.16, t_y^{(1)} = 0.11, t_x^{(2)} = 0.22, t_y^{(2)} = -0.16$), while still fitting the model with only one $U$. Second, we breached the assumption that the non-zero effects come from a Gaussian distribution. By design, the first three moments of the direct effects are fixed: they have zero mean, their variance is defined by the direct heritabilities and they must have zero skewness because the effect size distribution has to be symmetrical. Therefore, to violate the normality assumption, we varied the kurtosis (2, 3, 5 and 10) of the distribution drawn from the Pearson's distribution family. Third, we tested the assumption of the direct effects on our traits coming from a two-component Gaussian mixture by introducing a third component and observing how the estimates were effected. In this simulation scenario we introduced a large effect third component for $X$ while decreasing the polygenicity of $U$ ($\pi_{x1} = 1 \times 10^{-4}, \pi_{x2} = 1 \times 10^{-2}, h_{x1}^2 = 0.15, h_{x2}^2 = 0.1, \pi_u = 1 \times 10^{-2}$).

**Application to real summary statistics**. Once we demonstrated favourable performance of our method on simulated data, we went on to apply LHC-MR to summary statics obtained from the UK Biobank and other meta-analytic studies (Supplementary Table 1) in order to estimate pairwise bi-directional causal effect between 13 complex traits. The traits varied between conventional risk factors (such as low education, high body mass index (BMI), dislipidemia) and diseases (including diabetes and coronary artery disease among others). SNPs with imputation quality greater than 0.99, and minor allele frequency (MAF) greater than

0.5% were selected. Moreover, SNPs found within the human leukocyte antigen (HLA) region on chromosome 6 were removed due to the abundance of SNPs associated with autoimmune and infectious diseases as well as the complicated LD structure present in that region. For traits with total heritability below 2.5%, the outgoing causal effect estimates were ignored since instrumenting such barely heritable traits is questionable.

In order to perform LHC-MR between trait pairs, a set of overlapping SNPs was used as input for each pair. The effects of these overlapping SNPs were then aligned to the same effect allele in both traits. To decrease computation time further (while only minimally reducing power), we selected every 10th QC-filtered SNP as input for the analysis. We calculated regression weights using the UK10K panel, which may be sub-optimal for summary statistics not coming from the UK Biobank, but we have previously shown[47] that estimating LD in a ten-times larger dataset (UK10K) outweighs the benefit of using smaller, but possibly better-matched European panel (1000 Genomes[48]).

We also ran LHC-MR for each trait pair in both directions to estimate bi-directional causal effects as well as LD-score regression to get the cross-trait intercept term. We then added uniformly distributed ($\sim U(-0.1, 0.1)$) noise to these pre-estimated parameters to generate starting points for the second step of the likelihood optimisation. These closer-to-target starting points did not change the optimisation results, simply sped up the likelihood maximisation and increased the chances to converge to the same (primary) optimum. The LHC-MR procedure was run for each pair of traits 100 times, each using a different set of randomly generated starting points within the ranges of their respective parameters. For the optimisation of the likelihood function (Eq. (25)), we used the R function 'optim' from the 'stats' R package[49]. Once we fitted this *complete* model estimating 11 parameters in two steps $\{i_x, i_y, \pi_x, \pi_y, h_x^2, h_y^2, t_x, t_y, \alpha_{x \to y}, \alpha_{y \to x}, i_{xy}\}$, we then ran block jackknife to obtain the SE of the parameters estimated in the second step: $\{h_x^2, h_y^2, t_x, t_y, \alpha_{x \to y}, \alpha_{y \to x}, i_{xy}\}$.

To support the existence of the confounders identified by LHC-MR, we used EpiGraphDB[33,34] to systematically identify those potential confounders. The database provided for each potential confounder of a causal relationship, a causal effect on trait $X$ and $Y$ ($r1$, and $r3$ in their notation), the sign of the ratio of which ($sign(r_3/r_1)$) was compared to the sign of the LHC-MR estimated $t_y/t_x$ values representing the strength of the confounder acting on the two traits. We restricted our comparison to the sign only, since the $r1, r3$ values reported in EpiGraphDB are not necessarily on the same scale.

**Comparison against conventional MR methods and CAUSE**. We compared the causal parameter estimates of the LHC-MR method to those of five conventional MR approaches (MR-Egger, weighted median, IVW, mode MR and weighted mode MR) using a Z-test[50]. The 'TwoSampleMR' R package[51] was used to get the causal estimates for all the pairwise traits as well as their standard errors from the above-mentioned MR methods. The same set of genome-wide SNPs that were used by LHC-MR, were used as input for the package. SNPs associated with the exposure were selected to various degrees (for simulation we selected SNPs over a range of thresholds: absolute $p$-value $< 5 \times 10^{-4}$ to $< 5 \times 10^{-8}$), and SNPs more strongly associated with the outcome than with the exposure ($p$-value $< 0.05$ in one-sided $t$-test) were removed. The default package settings for the clumping of SNPs ($r^2 = 0.001$) were used and the analysis was run with no further changes. We tested the agreement between the significance and direction of our estimates and that of standard MR methods, with the focus being on finding differences in statistical conclusions regarding causal effect sizes.

We compared our causal estimates from all our simulation settings to the causal estimates obtained by running MR-RAPS[11] also using the 'TwoSampleMR' R package, once by using the entire set of SNPs, and another by filtering for SNPs with a significance threshold of $<5 \times 10^{-4}$. We also compared both our simulation as well as real data results against those of CAUSE[10]. We first generated simulated data under the LHC model and used them as input to estimate the causal effect using CAUSE. We then generated simulated data using the CAUSE framework and inputted them into LHC-MR (as well as standard MR methods) to estimate the causal parameters. Lastly, we compared causal estimates obtained for the 78 trait pairs (156 bi-directional causal effects) from LHC-MR to those obtained when running CAUSE.

**Reporting summary**. Further information on research design is available in the Nature Research Reporting Summary linked to this article.

## Data availability

The origin of the summary statistics data used is referenced in Supplementary Table 1. The UK Biobank summary statistics data used in this study came from the Neale Lab[52], and can be downloaded from http://www.nealelab.is/uk-biobank. Data on coronary artery disease[53] have been contributed by the CARDIoGRAMplusC4D and UK Biobank CardioMetabolic Consortium CHD working group who used the UK Biobank Resource (application number 9922). Data have been downloaded from http://www.cardiogramplusc4d.org/data-downloads/. The computed local LD scores described in Supplementary Methods 1.3 can be downloaded from https://wp.unil.ch/sgg/lhc-mr/. We also used EpiGraphDB, an analytical platform and database to support data mining

in epidemiology, to preform Phenome-wide MR search. Access to EpigraphDB is free and may be done through their web application https://epigraphdb.org or their R package https://github.com/MRCIEU/epigraphdb-r.

## Code availability

The source code[54] for this work can be found on https://github.com/LizaDarrous/LHC-MR_v2/(https://doi.org/10.5281/zenodo.5534639), it has also been developed into an R package that can be downloaded from https://github.com/LizaDarrous/lhcMR.

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

## Acknowledgements

This research has been conducted using the UK Biobank Resource under Application Number 16389. L.D. scores were calculated based on the UK10K data resource (EGAD00001000740, EGAD00001000741). Z.K. was funded by the Swiss National Science Foundation (31003A_169929, 310030_189147 and 32003B_173092). For computations we used the CHUV HPC cluster. We would like to thank Jack Bowden, Valentin Rousson, Matthew Robinson, George Davey Smith, Thomas Richardson, Eleonora Porcu and Sven Erik Ojavee for their valuable feedback and comments on this manuscript.

## Author contributions

Z.K. devised and directed the project. Z.K., N.M. and L.D. contributed to the mathematical derivations, design and implementation of the research, the analysis of the results and to the writing of the manuscript.

## Competing interests

The authors declare no competing interests.
