## [Peer Review File · Nature Communications]

Reviewers' Comments:

Reviewer #1:

Remarks to the Author:

The manuscript by Kutalik and colleagues proposed a new method for Mendelian Randomization (MR). It is motivated by the observation that latent heritable confounder could lead to biased estimates of causal effects by standard MR methods. It also has the advantage of using genomewide genetic variants. This method, LHC-MR (will be abbreviated as LHC in the review) is shown to improve parameter estimates in simulations, compared with standard MR methods. In real data, the results are largely consistent with standard MR, with some interesting differences.

The method is conceptually similar to the recently published method, CAUSE, as commented by the authors. The biggest difference seems to be how the two methods make inference of the parameters. While CAUSE uses a Bayesian strategy and focuses on model selection, LHC uses maximum likelihood estimation (MLE) and test parameters by LRT. However, the likelihood approach poses serious identifiability problems, where two solutions may exist with equally good likelihood. This point was commented by the authors, but is largely unaddressed. Below are specific comments:

Main comments

1) The likelihood function is not generally identifiable. This would lead to bimodal likelihood function as the authors pointed out. We think this is a fundamental weakness. We will first describe an important special case where it is easy to see how the problem happens and then show a more formal argument. We consider the case of "pure causal effect", i.e. $\alpha \neq 0$ (for simplicity, we consider X to Y effect only) and $q_y/q_x = 0$. Under this scenario, a fraction of π_x variants will have correlated effect sizes between Y and X, with the ratio equal to α , and the other variants do not show correlation. Now consider a model with only the confounder: $\alpha' = 0$, $\pi_u' = \pi_x$, $q_y'/q_x' = \alpha$ (the ' symbol denotes the parameters of the new model), and $h_u' = h_x$. It is easy to see that the pattern of data would be the same as before: we simply explain correlations due to causal effect via the confounder. This case is important because it is the model underlying standard MR methods. Having an identifiability problem with this simple case seems like a major weakness.

We can show that likelihood function is not identifiable in the general case. For simplicity, we set Y to X effect at 0. We write Σ_u matrix as: $h_u^2 q_x^2 / (m \pi_u)$, multiplied by a matrix whose diagonal terms are 1 and $(\alpha + q_y/q_x)^2$, and the off-diagonal term $(\alpha + q_y/q_x)$. Now it is easy to see that the likelihood function is symmetric with respect to variables u and x . So for any given model parameters, we have another model with exactly the same likelihood: the parameters of the new model are given by $\pi_x' = \pi_u$, $\alpha' = \alpha + q_y/q_x$, $h_x' = h_u$ and $\pi_u' = \pi_x$, $q_y'/q_x' = -q_y/q_x$, $h_u' = h_x$. It's not hard to understand intuitively what this transformation does. Under both models, the patterns of data would be the same: π_x percent of variants show correlation of effect sizes (between Y and X) α , and π_u percent of variants show correlation $\alpha + q_y/q_x$. In practice, one mode may be missed by LHC if $\alpha + q_y/q_x$ is outside their bound of parameters.

We have attached a small code example demonstrating this phenomenon. We view this as a fundamental problem because in many circumstances there may be two equally good solutions that imply very different sizes of causal effects. It is possible that in the simulations, the parameters happen to be chosen so that often a single mode is found (the other mode is outside the parameter range). However, in real data and in other simulations (see the comparison with CAUSE), the problem could occur much more often. Additionally, there is concern that the optimization algorithm may not always find both modes, even when both are within the ranges. So there is the risk of getting the false impression that there is only one solution.

2) Also regarding the identifiability problem: the authors described two procedures of testing effects using LRT or jackknife and show that they are similar in simulations. In real data though, the bimodal case is more common, and the authors chose to use the LRT test. However, it's

unclear when there are two modes, which one LHC would choose. What's the guideline here? More importantly, choosing one mode ignores a large amount of uncertainty in the data, and would inflate the statistical significance of results.

3) One nice aspect of LHC is that LD is explicitly accounted for during the model derivation. However, while LHC obtains the correct marginal distribution of summary statistics for a single variant, the statistics of nearby variants are not independent due to LD. So the likelihood function is still "incorrect" in that sense. What does LHC actually do in practice? Does it use LD pruning to get approximately independent SNPs?

4) It is not clear if the simulation setting is realistic. First of all, does it use real genotype data (with realistic LD) or use independent SNPs? It seems to be the latter, given that only 50K SNPs are used. While this is not atypical in this type of work, it does sound a bit unexpected given the careful treatment of LD earlier. Secondly, the proportion of variants associated with traits are quite high with $\pi_x = 0.1$ and $\pi_y = 0.15$. These numbers may not sound that high, if it includes SNPs in LD with causal variants. However, this may not be the case as commented earlier. Lastly, the default simulation setting uses $N = 500K$, which is very large. How many independent significant loci do trait X or Y have? Are they realistic?

5) Simulations only assess a single causal effect, with two possible values of the confounder effect. However, there may be many other scenarios where LHC will have trouble, as commented above in 1). In particular, a significant concern is whether the method will report causal effect when there is none. In the section comparing with CAUSE: it seems that this would happen quite often (Fig. S15, top panel). So it makes sense to add this "no causal effect" scenario to the main results (of course, it doesn't have to follow the CAUSE model to simulate such scenarios). Another scenario that could be important to study is: there is a causal effect, but the confounder effect (q_y/q_x) has an opposite sign.

6) Although the method is designed for bidirectional effects, they actually end up assuming that one of the effects is zero or the product of the two effects is very small. How was LHC applied in real data? In Fig. 3, were the estimated effects for a pair (in both directions) obtained from a single run using bi-directional model, or from two separate runs assuming one direction has zero effect? It should be described clearly. In any case, it would be good to explicitly demonstrate or comment on the possible advantage of including causal effects in both directions. A good example may be DM and BMI.

7) LHC is compared with CAUSE in simulations. When simulated under the LHC model, in the simple setting with only causal effect without a confounder, CAUSE was found to have only 3% power. Given the very large sample size, this is surprising. What is the power of LHC and other MR methods? When there is only causal effect, LHC and CAUSE models are almost identical. Could this be due to some problem of running CAUSE? Would it be possible that the results depend on some parameters of CAUSE?

8) Also about comparison with CAUSE in real data: only the pairs where all three methods (CAUSE, LHC and IVW) show significant findings were reported. It would be interesting to report more complete results, either as a supplementary table or figure like Fig. 4.

Minor comments

- It would be helpful to release the code as a package. What is available is a set of scripts that replicate the results in the paper. It is not easy to apply to new data because there are data specific parameters embedded in the code and it is fairly opaque.

- Fig. 2: smaller sample size setting, uses $n = 50K$. However, the text says $n = 10K$.

- It would be helpful to number the equations, especially the important ones.

- Line 681-682: it says that LHC is robust even when $\pi_u = 1$, and CAUSE has a significant false positive rate when $\pi_u > 0.3$. This does not seem right. First, the parameter q in

CAUSE is more like $\pi_u / (\pi_u + \pi_x)$, rather than π_u itself. Secondly, from the simulation under the CAUSE model, LHC has a serious problem even under $q = 0.1$: it identifies causal effects in 31% of time (Fig. S15).

- Writing can be improved. Some places feel long and wordy. Indention is not universally applied at the beginning of paragraphs (see the last second and their paragraphs of Results). Some paragraphs lack context or connections with preceding paragraphs: e.g. the last paragraph in the Results section.

```
library(ggplot2)
library(NMOF)
library(mvtnorm)
library(stats)
library(TwoSampleMR)

## Adapted from simulation code provided
## This code demonstrates that the likelihood is not identifiable.
## We identify two alternate sets of parameters. We then show that
## ll(Data | params1) = ll(Data  | params2)
## even though the two sets of parameters imply very different sizes of
causal effect.

## This is simulation function copied from DataSim.R

data_SIM <- function(pX = 0.1,
                     pU = 0.1,
                     pY = 0.1,
                     h2X = 0.25,
                     h2Y = 0.2,
                     h2U = 0.3,
                     qX = 0.3,
                     qY = 0.2,
                     a = 0.3,
                     b = 0.1,
                     M = 1e4,
                     nX = 1e6,
                     nY = 1e6,
                     NormDist = TRUE,
                     df = 5) {

  tX = qX * sqrt(h2U)
  tY = qY * sqrt(h2U)
  #tot_h2X = h2X + tX^2 + (b^2 * h2Y)
  #tot_h2Y = h2Y + tY^2 + (a^2 * h2X)

  if (NormDist) {
    ## normal
    piX = as.integer(runif(M, 0, 1) < pX) * (sqrt(h2X / (M * pX)) *
rnorm(M))
    piU = as.integer(runif(M, 0, 1) < pU) * (sqrt(h2U / (M * pU)) *
rnorm(M))
    piY = as.integer(runif(M, 0, 1) < pY) * (sqrt(h2Y / (M * pY)) *
rnorm(M))
    piNx = sqrt(1 / nX) * rnorm(M) #Error term
    piNy = sqrt(1 / nY) * rnorm(M) #Error term
  } else{
    ## Student
    piX = as.integer(runif(M, 0, 1) < pX) * (sqrt(h2X / (M * pX)) * rt(M,
df=df) /
                                                sqrt(df / (df - 2)))
    piU = as.integer(runif(M, 0, 1) < pU) * (sqrt(h2U / (M * pU)) * rt(M,
df=df) /
                                                sqrt(df / (df - 2)))
```

```

    piY = as.integer(runif(M, 0, 1) < pY) * (sqrt(h2Y / (M * pY)) * rt(M,
df=df) /
                                sqrt(df / (df - 2)))
    piNx = sqrt(1 / nX) * rnorm(M) #Error term
    piNy = sqrt(1 / nY) * rnorm(M) #Error term
}

#####

bX = piX + (qX + (b * qY)) * piU + (b * piY) + piNx
bY = piY + (qY + (a * qX)) * piU + (a * piX) + piNy

zX = sqrt(nX) * (bX) / sqrt(1 - bX ^ 2)
zY = sqrt(nY) * (bY) / sqrt(1 - bY ^ 2)

summary_list <-
  list(
    "bX" = bX,
    "bY" = bY,
    "zX" = zX,
    "zY" = zY
  )

return(summary_list)
}

```

```

## This function is coppied from the simulation code
### Structural Equation Model, calculates log likelihood from parameters
LHC_SEM = function(x) {

```

```

  pX = x[1]
  pU = x[2]
  pY = x[3]
  h2X = x[4]
  h2Y = x[5]
  tX = x[6]
  tY = x[7]
  a = x[8]
  b = x[9]

  L = NA

  S0 = matrix(c(1/nX, 0, 0, 1/nY), ncol = 2, nrow = 2)
  S1 = (h2X / (M*pX)) * matrix(c(1, a, a, a^2), ncol = 2, nrow = 2)
  S2 = (1 / (M*pU)) * matrix(c((tX + (b*tY))^2, (tX + (b*tY)) *
                                (tY + (a*tX))), (tX + (b*tY)) * (tY +
(a*tX)), (tY + (a*tX))^2), ncol = 2, nrow = 2)
  S3 = (h2Y / (M*pY)) * matrix(c(b^2, b, b, 1), ncol = 2, nrow = 2)

  mu = c(0,0)

  L = (pX*(1-pU)*(1-pY)*dmvnorm(D, mu, S1+S0))+
    (pU*(1-pX)*(1-pY)*dmvnorm(D, mu, S2+S0))+
    (pY*(1-pX)*(1-pU)*dmvnorm(D, mu, S3+S0))+
    (pX*pU*(1-pY)*dmvnorm(D, mu, S1+S2+S0))+

```

```

    (pX*pY*(1-pU)*dmvnorm(D, mu, S1+S3+S0))+
    (pY*pU*(1-pX)*dmvnorm(D, mu, S3+S2+S0))+
    (pY*pU*pX*dmvnorm(D, mu, S1+S2+S3+S0))+
    (1-pX)*(1-pU)*(1-pY)*dmvnorm(D, mu, S0)
  L[which(L==0)]=10e-100
  L2 = -sum(log(L))
  return(L2)
}

# Here is a set of parameters with a_yx = 0 and a_xy = 0.3
true_params <- list("pX"=0.1,"pY"=0.15, "pU"=0.05, "h2X"=0.25, "h2Y"=0.2,
  "h2U" = 0.3, "a"=0.3, "b"=0, qX = 0.3, qY = 0.1)
true_params$tX <- with(true_params, sqrt(h2U)*qX)
true_params$tY <- with(true_params, sqrt(h2U)*qY)
true_params$w <- with(true_params, qY/qX)

# This alternate set of parameters gives the same likelihood value:
alt_params <- with(true_params,
  list("pX"=pU,"pY"=pY, "pU"=pX, "h2X"=tX^2, "h2Y"=h2Y,
  "tX"=sqrt(h2X), "tY"=-w*sqrt(h2X), "a"=a + w, "b"=0))
alt_params$w <- with(alt_params, tY/tX)
alt_params$h2U <- 0.3
alt_params$qX <- with(alt_params, tX/sqrt(h2U))
alt_params$qY <- with(alt_params, tY/sqrt(h2U))

M = 5e4; nX = 5e5; nY = 5e5
NormDist = TRUE

# Simulate data from true_params

D_list = with(true_params, data_SIM(
  pX = pX, pU = pU, pY = pY,
  h2X = h2X, h2Y = h2Y, h2U = h2U,
  qX = qX, qY = qY, a = a, b = b,
  M = M, nX = nX, nY = nY,
  NormDist = TRUE
))
bX = D_list$bX #Effects of trait X
bY = D_list$bY #Effects of trait X
zX=D_list$zX; zY=D_list$zY #Standardised effects of traits X and Y
D = cbind(bX, bY)

# Likelihood with true_params and alt_params is the same
true_ll <- with(true_params, LHC_SEM(c(pX, pU, pY, h2X, h2Y, tX, tY, a,
b)))
alt_ll <- with(alt_params, LHC_SEM(c(pX, pU, pY, h2X, h2Y, tX, tY, a, b)))
true_ll == alt_ll

```

Reviewer #2:

Remarks to the Author:

Dear editor and authors,

Thank you for the opportunity to revise this contribution proposing a new Mendelian randomisation method -LHC-MR, which has the great advantage of using genome-wide information rather than a few selected instruments. This method is part of exciting developments in the field using genome-wide information for causal inference. The method simultaneously estimate bi-directional causal effects as well as explicitly accounts for confounding. This method looks extremely promising, with smaller bias and better efficiency than other MR methods across a number of simulations.

The paper is very comprehensive and overall clearly written, and I have few comments.

Methods

The authors have comprehensively compared their approach to several MR methods. I think it would make sense to also include MR-RAPS. this is because MR-RAPS is tailored for weak instruments and can therefore include variants at a more liberal threshold. it is therefore more comparable to LHC-MR in the sense that it can include more than genome-wide SNPs. It may be worth adding MR-RAPS to the mix, with the more lenient threshold proposed by authors of that method.

"estimate the impact of an infinite loop ... equilibrium state (i.e. converged variables)" can the authors provide a reference for this 'equilibrium state' they are referring to.

Is it possible to to justify further the choice of effect sizes for simulation parameters. In particular, I understand that you would want to to have only one causal effect to simplify the presentation, but why 0.3? This is a really big causal effect. Are findings similar with smaller causal effects? Similarly, how justifiable are effect sizes for the latent confounder.

The empirical example is based on the UK biobank. Usually, MR analysis is performed on summary statistics derived from two samples. This has consequences on estimates, for example the weak instrument bias is bias towards the observed association in one sample and towards the null in two sample MR. Can the authors comment on how the choice of focusing on one sample might affect findings.

Authors selected only a "10th QC-filtered SNP" as input for the analysis 'minimally reducing power'. Can this be explained a bit more, ie how selecting only a 10th of SNP will affect power only a little? Also, were these SNPs selected at random or as the most predicting SNPs.

In "The association summary statistic for SNP k of trait X can then be written as $B_{Xk} = g'_{k.X}/n_X$ " Can you please clarify for the reader how you come to that expression in the first place and the signification of the single quote ' in $g'_{k.X}$

Interpretation

" $\alpha_{X \rightarrow Y}$ and $\alpha_{Y \rightarrow X}$ in our model represent the total causal effects, some of which may be mediated by U ". Does this mean that U may be partly a mediator of reciprocal causal effects? What does this mean for the interpretation of the confounding effect as a percentage of the genetic correlation. I assume that this percentage still only include confounding effects?

Can the role of model selection be explained in simple terms. Namely, is the best fitting model simply there for information or are the final causal parameters estimated from the best fitting model? In SEM fitting, it is often the case that one parameter is not significant, but removing it can bias significantly other parameters (e.g. removing the C component in ACE twin models will inflate A). Here, if confounding is found to be not significant, will causal estimates from the model without confounding be biased compared to the same causal estimates from the complete model?

Do the authors have a recommendation on how to deal with the problem of two local maxima? For example, would interpreting LHC estimate in conjunction with other MR estimates help? It seems that the second local maximum is much further from other MR estimates.

Discussion

Is there any justification for choosing a minimum threshold of 1% for direct heritability.

At the end of the the discussion, a number of scenarios where LHC is not recommended are presented. I suggest that a small summary table of when to use LHC or when to use preferentially other MR methods would be useful, e.g. direct heritability > 1%, very large SNPs (use LHC but filter out large effect variants). This will help readers to make clear decisions re: when to apply or not the method.

In fig S6, it appears that when the sample size for X is much bigger than sample size for Y, there are a lot of outliers in the estimates from LHC. When sample size for Y > than for X, then we observe outliers for MR Egger Instead. Any recommendation on when not to use LHC (eg small sample size or unbalanced sample size can be added to the previous table)..

Writing.

In the introduction, please change the sentence " it is easy to see". this paper should be read by a broad audience in Nature Communications, including some readers with less technical background for whom it will not be easy to see. in general, when preparing the manuscript for publication, please assume as little knowledge as possible from your reader.

And the beginning of the introduction, the paper would benefit from a lesser or more cautious use of adjectives:

- 'modern day risk factors'? how is high blood pressure 'modern'? I think 'modern' can be removed safely.

- "due to its glaring abundance." experiments are quite abundant too

- adjectives such as "burning" or 'remarkable' do not add anything to the understanding of the topic and make the writing less to the point

- "casual" instead of "causal" in some places

Reviewer #3:

Remarks to the Author:

This paper proposes a new MR method, LHC-MR, to estimate the causal effect of a heritable risk factor. The paper aims to solve two very challenging issues in MR: one is estimating the direction of causal effects especially when the risk factor and the disease can causally affect each other; the other is horizontal pleiotropy, especially when the INSIDE assumption is violated. However, though the paper claims to solve the two issues, based on my understanding, I do not think it is able to solve either of them with their model described in the paper. Here are my main comments:

1. The structural equations after line 82, which is the fundamental model used in the paper, are problematic. First, they are not obviously valid structure equations. The paper explained in briefly in line 87 that "we however, estimate the impact of an infinite loop and model with the equilibrium state (i.e. converged variables)" which sounds intuitive, it is still hard to think how the data can be generated from the structural equations. Can the authors write down what exactly are the temporal forms of these equations, what the "equilibrium state" exactly means and how they converge to the current equations that are in the paper?

Second, even if these are valid structure equations, it is not able to identify the causal effects $\alpha_{y \rightarrow x}$ and $\alpha_{x \rightarrow y}$ from the data (even assuming that the confounder U is observed). The model with the first two equations is a special case of the simultaneous equation models (assuming U is observed), and the parameters $\alpha_{y \rightarrow x}$ and $\alpha_{x \rightarrow y}$ are not identifiable without further assumptions (see Wikipedia: Simultaneous equations model).

Third, this model with the priors assumed in the paper do not go further beyond the INSIDE

assumption for the hidden confounders. The horizontal pleiotropy ($\gamma_u q_y + \gamma_y$) are assumed to have mean 0 and are i.i.d. across SNPs, which is essentially the same assumption as INSIDE.

Finally, are e_X , e_Y and e_U assumed to be independent with each other? For the assumptions after line 99, κ_x is a vector for m genes, are the elements assumed to be i.i.d. following the same normal distribution? The model written in the paper is not complete without these specifications.

So because of the problems in these structural equations, I do not think the paper can either solve the causal directionality issue nor the confounding beyond INSIDE issue.

2. I think the derivations on pages 4-7 need to be rewritten. They are really sloppy, hard to follow and contain mathematical errors. Specially, there are three places that I want to point out. First, at line 135, the authors introduced the simplification $1 - \alpha_{y \rightarrow x} \alpha_{x \rightarrow y} \approx 1$. Do the authors assume that either $\alpha_{y \rightarrow x} = 0$ or $\alpha_{x \rightarrow y} = 0$? If not, what does that approximation exactly mean? If yes, then how is the method proposed in the paper better than other bi-directional MR approaches?

Second, line 142, I find the definition of π_x , π_y and π_u confusing. I think there is a typo in line 142 where $\rho_k \odot \zeta$ is not well-defined. It should be either $\rho_k \odot \zeta$, which is an m dimensional vector, or $\rho_k \zeta$ which is a scalar. Based on the assumptions after line 99, as ζ_x , ζ_y , ζ_u are random, the π_x , π_y , π_u are also random as they are functions of ζ_x , ζ_y , ζ_u , then how can they become parameters in the likelihood (after line 169)?

Third, as one can already see in the second issue, π_x , π_y , π_u are random variables. It is apparent that the distributions derived after line 151 are incorrect. Can the authors provide more mathematical details on how they find these to be two-component Gaussian mixtures? They are summations of Gaussian mixtures, so I don't think these results are mathematically correct.

1034 Reviewer Comments

1035 Overall responses: We are grateful for the insightful comments of all reviewers. They have
1036 led to significant improvements of the original work, in particular the distinction between two
1037 underlying models of effect sizes. In the old version of LHC-MR we assumed that the (true)
1038 marginal effects came from a spike and slab distribution with three major components (where
1039 the non-zero component is proportional to the LD score), X , Y and U -based. The most similar
1040 method, CAUSE, also assumed a distinct Gaussian mixture type architecture for the marginal
1041 effects (although they mention LD-transformed effects, but, to our understanding, they do not
1042 accordingly update the density function and keep working with Gaussians, assuming a single
1043 causal variant per locus, which is probably a reasonable assumption for most traits). While
1044 this is a reasonable assumption for both LHC-MR and CAUSE, it is not directly linked to an
1045 underlying multivariable genetic architecture, like the one assumed by LD Score Regression
1046 (LDSC). The authors of CAUSE argue that the number of causal markers tends to be less than
1047 independent LD blocks, but this is debatable since GWAS on complex traits including over a
1048 million sample size point to the fact that multiple independent causal effects exist for most of
1049 disease associated regions¹²³.

1050 To address this weakness, we have now modified LHC-MR to be rigorously derived from a **spike-**
1051 **and-slab multivariable causal SNP-effect model**. This results in a more complicated form
1052 of the likelihood function, but represents a truly unified model of the genetic architecture and
1053 causal relationships. The key difference between the old and the new LHC-MR model is that
1054 while the previous version assumed a spike-and-slab distribution of the **marginal effects of**
1055 **the GWAS SNPs** (which are contaminated by LD), the new version assumes a spike-and-slab
1056 **multivariable effect size distribution of the underlying causal variants** and from this
1057 we derived the distribution of the marginal effects.

1058 Reviewer #1 (Remarks to the Author):

1059 The manuscript by Kutalik and colleagues proposed a new method for Mendelian Random-
1060 ization (MR). It is motivated by the observation that latent heritable confounder could lead
1061 to biased estimates of causal effects by standard MR methods. It also has the advantage of
1062 using genome-wide genetic variants. This method, LHC-MR (will be abbreviated as LHC in
1063 the review) is shown to improve parameter estimates in simulations, compared with standard
1064 MR methods. In real data, the results are largely consistent with standard MR, with some
1065 interesting differences.

1066 We thank the reviewer for the positive comment.

1067 The method is conceptually similar to the recently published method, CAUSE, as commented
1068 by the authors. The biggest difference seems to be how the two methods make inference of the
1069 parameters. While CAUSE uses a Bayesian strategy and focuses on model selection, LHC uses
1070 maximum likelihood estimation (MLE) and test parameters by LRT. However, the likelihood
1071 approach poses serious identifiability problems, where two solutions may exist with equally good

¹Hormozdiari, F., Jung, J., Eskin, E. et al. MARS: leveraging allelic heterogeneity to increase power of association testing. *Genome Biol* 22, 128 (2021). <https://doi.org/10.1186/s13059-021-02353-8>

²Yang J, Ferreira T, Morris AP, et al. Conditional and joint multiple-SNP analysis of GWAS summary statistics identifies additional variants influencing complex traits. *Nat Genet.* 2012;44(4):369-S3. Published 2012 Mar 18. doi:10.1038/ng.2213

³Wood AR, Hernandez DG, Nalls MA, et al. Allelic heterogeneity and more detailed analyses of known loci explain additional phenotypic variation and reveal complex patterns of association. *Hum Mol Genet.* 2011;20(20):4082-4092. doi:10.1093/hmg/ddr328

1072 likelihood. This point was commented by the authors, but is largely unaddressed. Below are
1073 specific comments:

1074 Main comments

1075 1) The likelihood function is not generally identifiable. This would lead to bimodal likelihood
1076 function as the authors pointed out. We think this is a fundamental weakness. We will first
1077 describe an important special case where it is easy to see how the problem happens and then
1078 show a more formal argument. We consider the case of "pure causal effect", i.e $\alpha \neq 0$ (for
1079 simplicity, we consider X to Y effect only) and $q_y/q_x = 0$. Under this scenario, a fraction of
1080 π_x variants will have correlated effect sizes between Y and X , with the ratio equal to α , and
1081 the other variants do not show correlation. Now consider a model with only the confounder:
1082 $\alpha' = 0$, $\pi'_u = \pi_x$, $q'_y/q'_x = \alpha$ (the $'$ symbol denotes the parameters of the new model), and
1083 $h'_u = h_x$. It is easy to see that the pattern of data would be the same as before: we simply
1084 explain correlations due to causal effect via the confounder. This case is important because it is
1085 the model underlying standard MR methods. Having an identifiability problem with this simple
1086 case seems like a major weakness.

1087 We can show that likelihood function is not identifiable in the general case. For simplicity, we set
1088 Y to X effect at 0. We write Σ_u matrix as: $h_u^2 q_x^2 / (m\pi_u)$, multiplied by a matrix whose diagonal
1089 terms are 1 and $(\alpha + q_y/q_x)^2$, and the off-diagonal term $(\alpha + q_y/q_x)$. Now it is easy to see
1090 that the likelihood function is symmetric with respect to variables U and X . So for any given
1091 model parameters, we have another model with exactly the same likelihood: the parameters of
1092 the new model are given by $\pi'_x = \pi_u$, $\alpha' = \alpha + q_y/q_x$, $h'_x = h_u$ and $\pi'_u = \pi_x$, $q'_y q'_x = -q_y/q_x$,
1093 $h'_u = h_x$. It's not hard to understand intuitively what this transformation does. Under both
1094 models, the patterns of data would be the same: π_x , percent of variants show correlation of effect
1095 sizes (between Y and X) α , and π_u percent of variants show correlation $\alpha + q_y/q_x$. In practice,
1096 one mode may be missed by LHC if $\alpha + q_y/q_x$ is outside their bound of parameters.

1097 We have attached a small code example demonstrating this phenomenon. We view this as a
1098 fundamental problem because in many circumstances there may be two equally good solutions
1099 that imply very different sizes of causal effects. It is possible that in the simulations, the
1100 parameters happen to be chosen so that often a single mode is found (the other mode is outside
1101 the parameter range). However, in real data and in other simulations (see the comparison with
1102 CAUSE), the problem could occur much more often. Additionally, there is concern that the
1103 optimisation algorithm may not always find both modes, even when both are within the ranges.
1104 So there is the risk of getting the false impression that there is only one solution.

1105 Response #1: Thank you for this very constructive comment. This property was indeed expected
1106 based on the model, but we did not work out the exact conditions when two different set of
1107 parameters give identical likelihood values. This is a very logical property and translates into
1108 swapping direct with indirect effects, i.e. switching the roles of effects acting directly on X
1109 (γ_x) with effects that act through U (γ_u). Without the knowledge of U , these two scenarios
1110 can never be distinguished. This is not a weakness of the model or the method, but is a very
1111 realistic issue, which cannot be resolved by statistical means, instead we can only decide between
1112 the models guided by biological insights. Visually speaking, there are three slopes obtained from
1113 the observed $\widehat{\beta}_x$ - $\widehat{\beta}_y$ scatter plot: $\alpha_{x \rightarrow y}$, $\frac{\alpha_{x \rightarrow y} + \frac{q_y}{q_x}}{1 + \frac{q_y}{q_x} \cdot \alpha_{y \rightarrow x}}$ and $1/\alpha_{y \rightarrow x}$. Some of these can be confused
1114 with each other. By definition $|\alpha_{x \rightarrow y}| < 1$ and $|1/\alpha_{y \rightarrow x}| > 1$. If the third slope, $\frac{\alpha_{x \rightarrow y} + \frac{q_y}{q_x}}{1 + \frac{q_y}{q_x} \cdot \alpha_{y \rightarrow x}}$,
1115 is smaller than 1, it can be confused with $\alpha_{x \rightarrow y}$, otherwise it can be confused with $1/\alpha_{y \rightarrow x}$.
1116 Without loss of generality, let us assume that $\left| \frac{\alpha_{x \rightarrow y} + \frac{q_y}{q_x}}{1 + \frac{q_y}{q_x} \cdot \alpha_{y \rightarrow x}} \right| < 1$ and hence confused with $\alpha_{x \rightarrow y}$.

1117 In supplementary materials 1.1, we have derived the alternate parameters that could lead to the
1118 same likelihood solution. The special case of the solution (when $\alpha_{y \rightarrow x} = 0$) is what was kindly
1119 derived by the reviewer.

1120 Briefly, given that the SNPs effects between trait X and the confounder U are flipped, the new
1121 parameters have to satisfy the following equations:

$$\begin{aligned}h'_x &= t_x + t_y \cdot \alpha_{y \rightarrow x} \\h_x &= t'_x + t'_y \cdot \alpha'_{y \rightarrow x} \\ \alpha'_{x \rightarrow y} \cdot h'_x &= t_x \cdot \alpha_{x \rightarrow y} + t_y \\ \alpha_{x \rightarrow y} \cdot h_x &= t'_x \cdot \alpha'_{x \rightarrow y} + t'_y \\ h'_y &= h_y \\ \alpha'_{y \rightarrow x} &= \alpha_{y \rightarrow x}\end{aligned}$$

1122

1123 These two parameter settings give equal likelihood function values for the same data set.
1124 As a consequence of this reparameterisation, finding one optimum allows us to directly find the
1125 other one without the need to search for it. It can happen that the alternate set of parameters
1126 lies outside our parameter estimate constraints, leading to a unique solution of the likelihood
1127 minimisation. When both primary and alternative parameters are within range, the decision
1128 can only be based on biological knowledge. For example, it is reasonable to assume that the
1129 direct heritability of each trait is larger than the indirect heritability. (It is analogous to the
1130 prior distributions set out by CAUSE to avoid this issue.) Of course, a trait can have many
1131 indirect genetic effects, but most of those would not be shared with another trait. For these
1132 reasons, we prefer the parameter set where $h_x^2 > t_x^2$ or $h_y^2 > t_y^2$. Real data applications very
1133 consistently show that this decision leads to more biologically plausible solutions (i.e. results
1134 agreeing with epidemiological observations and smaller causal effects).

1135 2) Also regarding the identifiability problem: the authors described two procedures of testing
1136 effects using LRT or jackknife and show that they are similar in simulations. In real data though,
1137 the bimodal case is more common, and the authors chose to use the LRT test. However, it's
1138 unclear when there are two modes, which one LHC would choose. What's the guideline here?
1139 More importantly, choosing one mode ignores a large amount of uncertainty in the data, and
1140 would inflate the statistical significance of results.

1141 Response #2: This is a very keen observation, and we have thus changed the method and exclu-
1142 sively use block jackknife to estimate the standard errors (SEs) of our parameter estimations.
1143 Further expanding on our response to the first comment above, we use informative starting
1144 points to avoid bimodal effects. However we (can) calculate the alternate set of parameters ob-
1145 tained when the exposure or the outcome switches with the confounder. The block jackknife can
1146 provide SEs for each optimum. The choice between these two sets of parameters has to be based
1147 on our current understanding of the genomic architecture and additional independent evidence.
1148 Prompted by the reviewer, we have now included in the Discussion very explicit guidelines on
1149 how to choose between the two models:

- 1150 • For non-environmental traits the indirect heritability (that of the confounder) should be
1151 smaller than the direct heritability of the exposure (i.e. $h_x^2 > t_x^2$ and $h_y^2 > t_y^2$);
- 1152 • Since generally causal effect sizes are modest, we would assume that the optimum corre-
1153 sponding to smaller (in absolute value) causal effects is more reasonable.

1154 We calculated these alternate parameters for real data pairs that have sufficient heritability
1155 ($> 2.5\%$), and observed 47 cases where the alternate set resulting from switching the exposure

1156 and the confounder fall within our accepted ranges. Similarly, we found 55 cases where the
1157 alternate set is plausible when switching the outcome and the confounder. For all these cases
1158 however, the alternative set of parameters had direct heritability that was substantially smaller
1159 than the indirect heritability and also the causal effects were often unrealistically high (>0.5).
1160 This reassured us that the primary set of parameters are much more plausible and hence were
1161 reported in the main text. As for obtaining the SEs for our set of parameters, we use the
1162 estimated parameters with the maximum likelihood as a starting point set for our 200 block
1163 jackknife optimisation, thus obtaining CI specific to that optimum.

1164 3) One nice aspect of LHC is that LD is explicitly accounted for during the model derivation.
1165 However, while LHC obtains the correct marginal distribution of summary statistics for a single
1166 variant, the statistics of nearby variants are not independent due to LD. So the likelihood
1167 function is still "incorrect" in that sense. What does LHC actually do in practice? Does it use
1168 LD pruning to get approximately independent SNPs?

1169 Response #3: This is an excellent point and there are two ways around this problem. While
1170 CAUSE solves this via pruning the observed summary statistics, we chose to apply the same
1171 trick as LD score regression, namely weighting the likelihood function values. As described in the
1172 Methods (Eq. 2), instead of summing up the SNP-level log-likelihood values, $l(\theta) = \sum_k l_k(\theta)$, we
1173 use a down-weighted sum $l(\theta) = \sum_k \frac{1}{\sum_j r_{k,j}^2} \cdot l_k(\theta)$, where each log-likelihood term is multiplied
1174 by the inverse of the sum of squared LD between the focal variant and all other tested variants.
1175 What it does is that if for example there are five GWAS SNPs in perfect LD, their log-likelihood
1176 values will be divided by five, i.e. they will still only count for one marker.

1177 4) It is not clear if the simulation setting is realistic. First of all, does it use real genotype data
1178 (with realistic LD) or use independent SNPs? It seems to be the latter, given that only 50K
1179 SNPs are used. While this is not atypic in this type of work, it does sound a bit unexpected
1180 given the careful treatment of LD earlier. Secondly, the proportion of variants associated with
1181 traits are quite high with $\pi_x = 0.1$ and $\pi_y = 0.15$. These numbers may not sound that high, if
1182 it includes SNPs in LD with causal variants. However, this may not be the case as commented
1183 earlier. Lastly, the default simulation setting uses $N = 500K$, which is very large. How many
1184 independent significant loci do trait X or Y have? Are they realistic?

1185 Response #4: This is another excellent point. Simulation settings were too simplistic (maximum
1186 one causal effect per LD block). Instead, we now generate summary statistics based on real geno-
1187 type data (with marginal effect sizes being proportional to the LD score). As described in our
1188 overall response, we have now improved the model so that it does not rely on the assumption of a
1189 single causal variant per locus (see the updated Methods section). We have rerun our upgraded
1190 likelihood function for a wider set of simulation scenarios. We also agree with the reviewer that
1191 there are two types of polygenicity measures: previously we used π_x as the proportion of non-
1192 zero *marginal* effects, for which $\pi = 0.1$ was reasonable, but now our updated model defines π_x
1193 as the proportion of non-zero *multivariable* causal effects among all sequence variants (directly
1194 influencing X). Therefore, in full agreement with the reviewer, we set this value lower in the
1195 updated simulations. For real data applications we observed trait polygenicity values between
1196 10^{-5} and 10^{-4} , but as we saw in simulations LHC-MR can underestimate these parameters even
1197 up to five folds, thus real values can be as high as 10^{-3} . Hence, for the simulations, we chose
1198 polygenicity values between 5×10^{-3} up to 0.15 to cover all ranges. Finally, we have also ran all
1199 simulations for $N=500k$ and also $N=50K$, where we have independent significant SNPs ranging
1200 from 1300 – 2000 for larger sample sizes, and very few (0 – 2 SNPs) for smaller sample sizes,
1201 which are not unrealistic numbers.

1202 5) Simulations only assess a single causal effect, with two possible values of the confounder effect.

1203 However, there may be many other scenarios where LHC will have trouble, as commented above
1204 in 1). In particular, a significant concern is whether the method will report causal effect when
1205 there is none. In the section comparing with CAUSE: it seems that this would happen quite
1206 often (Fig. S15, top panel). So it makes sense to add this "no causal effect" scenario to the
1207 main results (of course, it doesn't have to follow the CAUSE model to simulate such scenarios).
1208 Another scenario that could be important to study is: there is a causal effect, but the confounder
1209 effect q_y/q_x has an opposite sign.

1210 Response #5: Thank you for this comment. We have added several simulation scenarios on
1211 top of those that were presented before. As suggested by the reviewer, these include a "no
1212 causal effect" ($\alpha_{y \rightarrow x} = 0$) and an "opposite confounder effect" ($q_x = 0.3, q_y = -0.2$) scenario,
1213 as well as a scenario with a smaller causal effect ($\alpha_{x \rightarrow y} = 0.1$) (Figures 2, S4, S6, S7). Under
1214 all these scenarios, LHC-MR outperforms the standard MR methods in estimating the causal
1215 effect.

1216 6) Although the method is designed for bidirectional effects, they actually end up assuming that
1217 one of the effects is zero or the product of the two effects is very small. How was LHC applied in
1218 real data? In Fig. 3, were the estimated effects for a pair (in both directions) obtained from a
1219 single run using bi-directional model, or from two separate runs assuming one direction has zero
1220 effect? It should be described clearly. In any case, it would be good to explicitly demonstrate
1221 or comment on the possible advantage of including causal effects in both directions. A good
1222 example may be DM and BMI.

1223 Response #6: As we have updated the likelihood function, for simplicity we got rid of this
1224 simplification, hence none of our results rely on this assumption and our method works well
1225 even when strong bi-directional effects exist (we added now one simulation setting explicitly
1226 addressing this point). Note however that the interpretation of all parameters becomes difficult
1227 in the case of strong bi-directional effects, which is very rare for real data applications (see
1228 Figure S1). To exemplify this, let us consider the notion of the heritability parameter (e.g.
1229 $h_x^2 + t_x^2$). This would not mean heritability any longer, because this needs to be divided by
1230 $(1 - \alpha_{x \rightarrow y} \cdot \alpha_{y \rightarrow x})$ to get the total heritability through the infinite looping of the genetic effects
1231 between X and Y . The same holds for the causal effects, which would also represent a quantity
1232 of no practical use. One interpretation is time dependent parameters (as time passes the causal
1233 effects and heritability change, because more and more loops take place). For these reasons, we
1234 think the interpretation of the model parameters in case of strong bi-directional causal effects
1235 is beyond the scope of this paper.

Figure S1: Simulation results under strong bidirectional causal effects These Raincloud boxplots^[28] represent the distribution of parameter estimates from 50 different data generations. For each generation, standard MR methods as well as our LHC-MR were used to estimate a causal effect. The true values of the parameters used in the data generations are represented by the blue dots/lines. A strong bidirectional causal effect is studied here with $\alpha_{x \rightarrow y} = 0.4$ and $\alpha_{y \rightarrow x} = 0.3$. **a** Results for smaller sample size of $n_x = n_y = 50,000$. **b** Results for larger sample size of $n_x = n_y = 500,000$.

1236 7) LHC is compared with CAUSE in simulations. When simulated under the LHC model, in the
 1237 simple setting with only causal effect without a confounder, CAUSE was found to have only 3%
 1238 power. Given the very large sample size, this is surprising. What is the power of LHC and other
 1239 MR methods? When there is only causal effect, LHC and CAUSE models are almost identical.
 1240 Could this be due to some problem of running CAUSE? Would it be possible that the results
 1241 depend on some parameters of CAUSE?

1242 **Response #7:** Thank you for pointing out this problem. Indeed, upon detailed correspondence
 1243 with the authors of the CAUSE paper, we were able to find out that we had been running the
 1244 simulation of CAUSE using LHC-MR simulated data while providing the wrong parameter for

1245 the CAUSE analysis. We have thus updated the analysis pipeline and regenerated these results
1246 as seen in Figures S14, S15. For smaller sample sizes and in the presence of a shared factor,
1247 the sharing model wins 100% of the time, in the presence of a causal effect or in the presence
1248 of both, the causal model wins 4% of the time instead. For larger sample sizes however, the
1249 correct model is selected 100% of the time for its respective scenario (causal model when both
1250 causal and confounder effects are present). The interpretation of the causal effect γ in both
1251 sample sizes however remains the same, slightly underestimated when it is present. In the case
1252 of the shared effect model, the parameter q is estimated at 0.05 which is half of the suggested
1253 equivalent of LHC-MR parameters $\frac{t_x^2}{t_x^2+h_x^2} = 0.093$ (as suggested in minor comment - response
1254 12), but is more accurately estimated when $n_x = n_y = 500,000$.

1255 Furthermore, the parameter η is also underestimated when compared to our true confounder
1256 effect ratio on the traits ($t_y/t_x = 0.66$). Nevertheless, differences should be expected, due to
1257 numerous differences between CAUSE and (even the old version of) LHC-MR, notably CAUSE
1258 assuming identical per SNP effect for both X and U , ignoring SNPs with direct effect on Y .

1259 8) Also about comparison with CAUSE in real data: only the pairs where all three methods
1260 (CAUSE, LHC and IVW) show significant findings were reported. It would be interesting to
1261 report more complete results, either as a supplementary table or figure like Fig. 4.

1262 Response #8: The complete table of results is in fact in the Supplementary Table S5. Thank
1263 you for pointing this out however, we have updated the caption of table S8 (which shows results
1264 that are significant to LHC-M and CAUSE only) to more clearly indicate where the full results
1265 can be found.

1266 **Minor comments**

1267 - It would be helpful to release the code as a package. What is available is a set of scripts that
1268 replicate the results in the paper. It is not easy to apply to new data because there are data
1269 specific parameters embedded in the code and it is fairly opaque.

1270 Response #9: Thank you for this suggestion. An R package is indeed in the works, which will
1271 be complete before publication. Until then however, the scripts have been updated to reflect the
1272 new method, documented more thoroughly, and reshaped. The scripts are easy to apply to new
1273 data, as there are no specific parameters in them (any parameters present should be updated to
1274 fit the data being used, such as sample size and SNP number).

1275 - Fig. 2: smaller sample size setting, uses $n = 50K$. However, the text says $n = 10K$.

1276 Response #10: Thank you for this comment, we have since updated many of our simulation
1277 scenarios and focus on two different sample sizes for most of our scenarios: 50,000 and 500,000.
1278

1279 - It would be helpful to number the equations, especially the important ones.

1280 Response #11: Thank you for this suggestion, we have indeed gone back and numbered some of
1281 the equations so that they are easier to reference. However, we felt that numbering is justified
1282 when we refer to an equation later, hence kept it to a minimum following this principle.

1283 - Line 681-682: it says that LHC is robust even when $\pi_u = 1$ and CAUSE has a significant false
1284 positive rate when $\pi_u > 0.3$. This does not seem right. First, the parameter q in CAUSE is
1285 more like $\pi_u/(\pi_u + \pi_x)$ rather than π_u itself. Secondly, from the simulation under the CAUSE
1286 model, LHC has a serious problem even under $q = 0.1$: it identifies causal effects in 31% of time
1287 (Fig. S15).

1288 Response #12: The parameter q in CAUSE is equivalent to LHC-MR parameter $\frac{t_x^2}{t_x^2+h_x^2}$, and

1289 fitting the standard parameter setting values to the above expression we see that it equates to
1290 0.093. We also performed simulations with more severe confounding, where the $\frac{t_x^2}{t_x^2+n_x^2}$ ratio was
1291 0.4, both exposure and outcome GWAS were well powered and the equivalent of $\eta (t_y/t_x)$ was
1292 set to $\frac{2}{3}$ (Figure S8), i.e. far more extreme than the settings in Fig 4a in [10]. Under these
1293 settings we also observe some overestimation of the causal effects, hence we toned down our
1294 statement.

1295 Furthermore, in the updated cases where we fitted LHC-MR to CAUSE-simulated data, as
1296 seen in Figures S16 and S17, LHC-MR performs well in estimating the causal effect with larger
1297 sample sizes in the three differing scenarios. Note that to speed up computation, we used a
1298 smaller number of SNPs as input (97k vs 230k previously used).

1299 - Writing can be improved. Some places feel long and wordy. Indentation is not universally
1300 applied at the beginning of paragraphs (see the last second and their paragraphs of Results).
1301 Some paragraphs lack context or connections with preceding paragraphs: e.g. the last paragraph
1302 in the Results section.

1303 Response #13: Thank you for noting all of these. We have eliminated indentation and hope
1304 that the paragraphs and general flow are more legible this way. We have thoroughly re-written
1305 the Results and Discussion sections following the updates that were done and based on reviewer
1306 comments.

1307 Reviewer #2 (Remarks to the Author):

1308 Dear editor and authors, Thank you for the opportunity to revise this contribution proposing
1309 a new Mendelian randomisation method -LHC-MR, which has the great advantage of using
1310 genome-wide information rather than a few selected instruments. This method is part of excit-
1311 ing developments in the field using genome-wide information for causal inference. The method
1312 simultaneously estimate bi-directional causal effects as well as explicitly accounts for confound-
1313 ing. This method looks extremely promising, with smaller bias and better efficiency than other
1314 MR methods across a number of simulations. The paper is very comprehensive and overall
1315 clearly written, and I have few comments.

1316 We would like to thank the reviewer for the very positive comments.

1317 Methods

1318 The authors have comprehensively compared their approach to several MR methods. I think
1319 it would make sense to also include MR-RAPS. this is because MR-RAPS is tailored for weak
1320 instruments and can therefore include variants at a more liberal threshold. it is therefore more
1321 comparable to LHC-MR in the sense that it can include more than genome-wide SNPs. It may
1322 be worth adding MR-RAPS to the mix, with the more lenient threshold proposed by authors of
1323 that method.

1324 Response #14: Thank you for this suggestion. We have compared MR-RAPS with LHC-MR
1325 and other standard MR methods for all our simulation scenarios, using two different p-value
1326 thresholds to select instruments (5×10^{-4} , as suggested by the authors, and 1, keeping all
1327 SNPs). For our standard settings scenario, for example, we observed that MR-RAPS causal
1328 effect estimate using moderately strong instruments was downward biased (probably because
1329 of winner's curse) whereas the estimate obtained using all SNPs was upward biased (Figure
1330 2).

1331 "estimate the impact of an infinite loop ... equilibrium state (i.e. converged variables)" can the
1332 authors provide a reference for this 'equilibrium state' they are referring to.

1333 Response #15: We agree that it was not well formulated and such analogues can be confusing.
1334 We have simply reformulated the original equations such that there is no recursivity, i.e. X is
1335 originally defined as a function of Y , but Y is defined as a function of X . This formulation can
1336 be rearranged such that in the equations for X there is no explicit dependence on Y and vice
1337 versa. We removed any reference to an equilibrium state and time dependent equations.

1338 Is it possible to to justify further the choice of effect sizes for simulation parameters. In par-
1339 ticular, I understand that you would want to to have only one causal effect to simplify the
1340 presentation, but why 0.3? This is a really big causal effect. Are findings similar with smaller
1341 causal effects? Similarly, how justifiable are effect sizes for the latent confounder.

1342 Response #16: We thank the reviewer for this point and agree that smaller causal effects need
1343 to be explored too. For this reason we added two further simulation settings along this direction:
1344 (1) We simply reduce the causal effect to $\alpha_{x \rightarrow y} = 0.1$, while leaving other parameters untouched
1345 (Figures S6 and S7). (2) We simulate data with no causal effect (Figures S6 and S7). We have
1346 also addressed the suggestion to include causal effects in both directions and to make it even
1347 more difficult, we assigned opposite signs, $\alpha_{x \rightarrow y} = 0.3, \alpha_{y \rightarrow x} = -0.2$ (Figures 2 and S4).

1348 Real data results have shown that confounder effects are typically smaller than our simulation
1349 values, hence we feel that our parameter choices are broadly justified. Still to fully address the
1350 reviewer's concern we have now also performed simulations with much larger confounder effects
1351 $q_x = 0.75, q_y = 0.5$ (Figure S8). In total we now explore 32 different parameter settings.

1352 The empirical example is based on the UK biobank. Usually, MR analysis is performed on
1353 summary statistics derived from two samples. This has consequences on estimates, for example
1354 the weak instrument bias is bias towards the observed association in one sample and towards
1355 the null in two sample MR. Can the authors comment on how the choice of focusing on one
1356 sample might affect findings.

1357 Response #17: Indeed, standard MR methods, such as IVW-MR, are designed to be used in
1358 independent samples. However, we expect the bias induced by sample overlap in very large
1359 samples to be minimal, especially when a stringent threshold is used to select instruments. We
1360 have a manuscript under preparation exactly on this topic⁴. In that work we have shown that
1361 – in most settings – no sample overlap introduces more bias in the causal effect estimation of
1362 classical MR methods, than full sample overlap, but in any case with exposure and outcome
1363 sample sizes > 100K, the bias due to sample overlap is barely noticeable. Please note that our
1364 approach, LHC-MR automatically takes sample overlap into account (through the error term’s
1365 covariance element), making our results more reliable. As a consequence, for real data results
1366 with (partially) overlapping samples, discrepancies between standard MR methods and LHC-
1367 MR can then be due to 3 reasons: 1) the existence of a genetic confounder, 2) the difference in
1368 SNPs used (genome-wide SNPs for LHC-MR vs strong instruments only for standard methods),
1369 3) sample overlap.

1370 Authors selected only a ”10th QC-filtered SNP” as input for the analysis ”minimally reducing
1371 power”. Can this be explained a bit more, ie how selecting only a 10th of SNP will affect power
1372 only a little? Also, were these SNPs selected at random or as the most predicting SNPs.

1373 Response #18: The SNPs were selected at random, where the original input of effects on the
1374 exposure and effects on the outcome were thinned by removing every 10th SNP (when ordered
1375 by physical position). To investigate the effect of randomly selecting every n th SNP on the
1376 power, we took a region of 1000 SNPs on chromosome one, and thinned it by taking every SNP,
1377 2nd, 3rd, 4th, 5th, 10th, 20th, 30th, 40th, and 50th SNP for the analysis and measured the
1378 number of effective SNPs (number of SNPs that explain 99.5% of the variance) that result for
1379 each level of thinning. We then took the ratio of effective SNPs from the entire 1000 SNP region
1380 to that at each level of thinning (Figure S2 below). The reason for the non-linear decline is
1381 that SNPs that are close tend to have much higher LD, thus the information lost by removing a
1382 SNP can be recovered by the information carried by SNPs that were kept and in high LD with
1383 the removed variant. We observed that when taking every 10th SNP, the ratio is 0.64, meaning
1384 that we are losing 36% of the information. Despite the very small effect this has on SEs (slight
1385 increase), it is allowing us to be ten times faster in our calculations.

⁴Mounier, N., Kutalik, Z. Correction for sample overlap, winner’s curse and weak instrument bias in two-sample Mendelian Randomization. bioRxiv (2021). <https://doi.org/10.1101/2021.03.26.437168>

Figure S2: Investigating the effect of thinning by every nth SNP on the number of effective tests. a Boxplots reporting the number of independent SNPs/effective tests after thinning the 1000 SNP region by every nth SNP. b The ratio of the number of effective SNPs when taking all SNPs to the number SNPs when thinning by taking every nth SNP.

1386 In "The association summary statistic for SNP k of trait X can then be written as $\widehat{\beta}_k^x = \mathbf{g}'_k \cdot \mathbf{x}/n_x$ "
 1387 Can you please clarify for the reader how you come to that expression in the first place and the
 1388 signification of the single quote ' in \mathbf{g}'_k

1389 Response #19: We have explained it more clearly in the manuscript: the marginal effect size
 1390 estimate for SNP k on trait X can then be written as $\widehat{\beta}_k^x = \mathbf{g}'_k \cdot \mathbf{x}/n_x$, which is as special case
 1391 of univariable standard normal linear regression when both the outcome and the predictor are
 1392 standardised to have zero mean and unit variance^[12]. We also explain now that the ' sign means
 1393 matrix transpose.

1394 Interpretation

1395 " $\alpha_{x \rightarrow y}$ and $\alpha_{y \rightarrow x}$ in our model represent the total causal effects, some of which may be mediated
 1396 by U ". Does this mean that U may be partly a mediator of reciprocal causal effects? What
 1397 does this mean for the interpretation of the confounding effect as a percentage of the genetic
 1398 correlation. I assume that this percentage still only include confounding effects?

1399 Response #20: This statement refers to the very unlikely situation that was described just
 1400 before the quoted sentence, when U is not only a confounder, but also a consequence of X or Y .
 1401 Such situation cannot be distinguished by the classical model situation (when X and Y have no
 1402 causal effects on U) because it is not identifiable, thus the interpretation of the results of any
 1403 method in such a situation may be wrong. As a consequence the interpretation of the causal
 1404 effects, and the contribution of the confounding factor to the genetic correlation are under the
 1405 assumption that such reverse ($X \rightarrow U$ and $Y \rightarrow U$) effects are null.

1406 Can the role of model selection be explained in simple terms. Namely, is the best fitting model
 1407 simply there for information or are the final causal parameters estimated from the best fitting
 1408 model? In SEM fitting, it is often the case that one parameter is not significant, but removing
 1409 it can bias significantly other parameters (e.g. removing the C component in ACE twin models
 1410 will inflate A). Here, if confounding is found to be not significant, will causal estimates from the
 1411 model without confounding be biased compared to the same causal estimates from the complete
 1412 model?

1413 **Response #21:** Thank you for this thought-provoking question. Indeed, previously, the final
 1414 causal parameters estimated came from the best fitting model, the one nested till all the pa-
 1415 rameters were significant according to the LRT. But following the reviewer’s comment, we do
 1416 not perform nesting in the updated manuscript and rather use the block jackknife procedure the
 1417 obtain SE estimates.

1418 To follow-up the reviewer’s point, we also investigated, for the real trait application, how much
 1419 causal estimates could change if they were re-estimated for a nested model without any con-
 1420 founder (in case the full model yielded insignificant confounder). For 61 such trait pairs, we
 1421 re-ran a new model where the underlying structure did not have a confounder acting on the
 1422 two traits. We plotted those causal estimates against the original estimates from the complete
 1423 model in Figure S3, and noted a considerable concordance in both values and significance status
 1424 between the two models.

Figure S3: Real data causal estimate comparison between models. Scatterplot showing the causal effect estimate ($\alpha_{x \rightarrow y}$) from the complete model on the x-axis and that of running the model with no confounder on the y-axis for trait pairs that did not have a significant confounder effect. The one-to-one line is shown, where most points lie. Changes to the causal estimates from the complete model to the no-confounder model are coloured, where most maintained their significance level. Only one trait pair had a statistically significant difference between the estimates of the two models (LDL \leftrightarrow DM). The other labelled pair has an estimate difference greater than 0.25.

1425 Do the authors have a recommendation on how to deal with the problem of two local maxima?
1426 For example, would interpreting LHC estimate in conjunction with other MR estimates help?
1427 It seems that the second local maximum is much further from other MR estimates.

1428 **Response #22:** As shown in comment and response #1, the identifiability of the likelihood
1429 function makes it so that there are two possible solutions when the direct and indirect genetic
1430 effects on the exposure (or the outcome) are swapped. We have now derived the analytical
1431 formula (see Supplementary Methods 1.1) how to obtain the alternative optimum once one of
1432 them is identified, hence we can never miss it, even if it is outside the parameter constraint box.
1433 The alternate set of parameter estimates could fall outside of our parameter estimate constraints
1434 (in this case we observe one mode) or inside the constraints to give us the two equally likely
1435 maxima and their corresponding parameter estimates.

1436 The choice to report one of the two is based on our current understanding of the genomic
1437 architecture and additional independent evidence. Prompted by the reviewer, we have now
1438 included in the Discussion suggested guidelines for choosing between two models:

- 1439 • For non-environmental traits the indirect heritability (that of the confounder) should be
1440 smaller than the direct heritability of the exposure (i.e. $h_x^2 > t_x^2$ and $h_y^2 > t_y^2$).
- 1441 • Since generally causal effect sizes are modest, we would assume that the optimum corre-
1442 sponding to smaller (in absolute value) causal effects is more reasonable.

1443 Discussion

1444 Is there any justification for choosing a minimum threshold of 1% for direct heritability.

1445 **Response #23:** The 1% direct heritability threshold is the limit of detection given the sample
1446 size. Taking the LDSC data that roughly shares the same sample size as our traits, and selecting
1447 the UK Biobank traits that are continuous and available for both sexes, we can calculate the
1448 minimum heritability needed for detection. Taking the ratio of the heritability to the SE, and
1449 selecting the traits that have a significant ratio above the Z-value that corresponds to a significant
1450 p-value of $0.05/(\text{number of pairs} * 2)$ (Z-value of 3.60), the smallest heritability found is 0.0109.
1451 We have since updated this threshold to 2.5% *total* heritability (instead of 1% *direct* heritability)
1452 to be more cautious of the causal estimates that we report. Only two traits have a total
1453 heritability below this threshold: Myocardial infarction (MI) and Simvastatin (SVstat).

1454 At the end of the the discussion, a number of scenarios where LHC is not recommended are
1455 presented. I suggest that a small summary table of when to use LHC or when to use preferentially
1456 other MR methods would be useful, e.g. direct heritability $> 1\%$, very large SNPs (use LHC
1457 but filter out large effect variants). This will help readers to make clear decisions re:when to
1458 apply or not the method.

1459 **Response #24:** That is a very sound suggestion, thank you. This has since changed with our
1460 improved method, which has far less limitations. We have listed them more clearly in the
1461 Discussion.

1462 In fig S6, it appears that when the sample size for X is much bigger than sample size for Y ,
1463 there are a lot of outliers in the estimates from LHC. When sample size for $Y >$ than for X ,
1464 then we observe outliers for MR Egger Instead. Any recommendation on when not to use LHC
1465 (eg small sample size or unbalanced sample size can be added to the previous table)..

1466 **Response #25:** Indeed when there is an imbalance in the sample size between the exposure and
1467 the outcome, the parameter estimations tend to be more biased when the exposure has a lower
1468 sample size. However, such settings impact the updated LHC-MR method far less than its older
1469 version or other MR methods, see Figure S5. Standard MR methods with smaller sample sizes

1470 will have less instruments for the causal estimation, thus less power. Interestingly, MR-RAPS
1471 is affected by the exposure having a smaller sample size than the outcome (as apposed to both
1472 having the same sample size in Figure 2).

1473 **Writing.** In the introduction, please change the sentence " it is easy to see". this paper
1474 should be read by a broad audience in Nature Communications, including some readers with
1475 less technical background for whom it will not be easy to see. in general, when preparing the
1476 manuscript for publication, please assume as little knowledge as possible from your reader.

1477 And the beginning of the introduction, the paper would benefit from a lesser or more cautious use
1478 of adjectives: -'modern day risk factors'? how is high blood pressure 'modern'? I think 'modern'
1479 can be removed safely. -"due to its glaring abundance." experiments are quite abundant too -
1480 adjectives such as 'burning' or 'remarkable' do not add anything to the understanding of the
1481 topic and make the writing less to the point -"casual" instead of "causal" in some places

1482 **Response #26:** Thank you for these comments, they are really helpful and constructive to us
1483 approaching a broader audience. We have re-worded the introduction and much of the text in
1484 general to reflect your suggestions.

1485 **Reviewer #3 (Remarks to the Author):**

1486 This paper proposes a new MR method, LHC-MR, to estimate the causal effect of a heritable
1487 risk factor. The paper aims to solve two very challenging issues in MR: one is estimating the
1488 direction of causal effects especially when the risk factor and the disease can causally affect each
1489 other; the other is horizontal pleiotropy, especially when the INSIDE assumption is violated.
1490 However, though the paper claims to solve the two issues, based on my understanding, I do not
1491 think it is able to solve either of them with their model described in the paper. Here are my
1492 main comments:

1493 1. The structural equations after line 82, which is the fundamental model used in the paper,
1494 are problematic. First, they are not obviously valid structure equations. The paper explained
1495 in briefly in line 87 that "we however, estimate the impact of an infinite loop and model with
1496 the equilibrium state (i.e. converged variables)" which sounds intuitive, it is still hard to think
1497 how the data can be generated from the structural equations. Can the authors write down what
1498 exactly are the temporal forms of these equations, what the "equilibrium state" exactly means
1499 and how they converge to the current equations that are in the paper?

1500 **Response #27:** Indeed, we did not describe it well and unnecessarily included analogues to
1501 temporal modelling and equilibrium states, which was more confusing than useful. To make the
1502 derivation clean, starting from the same original model equations, we turned them into non-
1503 recursive forms, i.e. where X and Y do not explicitly depend on each other. These equations
1504 also allow easy data generation. This is known as the *reduced form* in econometrics.

1505 Second, even if these are valid structure equations, it is not able to identify the causal effects
1506 $\alpha_{y \rightarrow x}$ and $\alpha_{x \rightarrow y}$ from the data (even assuming that the confounder U is observed). The model
1507 with the first two equations is a special case of the simultaneous equation models (assuming U is
1508 observed), and the parameters $\alpha_{y \rightarrow x}$ and $\alpha_{x \rightarrow y}$ are not identifiable without further assumptions
1509 (see Wikipedia: Simultaneous equations model).

1510 **Response #28:** In the following we show that while Simultaneous equation models (SimEq)
1511 bear similarity to our problem setting, it also has key differences. We will refer to notation on
1512 the wikipedia page to make sure there is no ambiguity. First, the major difference between our
1513 approach and that of SimEq is that the latter aims to estimate all parameters (Γ and B), in
1514 our case B is just a nuisance parameter and integrated out, as often done in Bayesian inference.
1515 Therefore, identifiability assumptions listed for SimEq are not relevant to our goal, since we do
1516 not aim to identify all parameters (we do not aim to identify any of the genetic effects).

1517 Following up on this point: We could make our problem more similar to SimEq, by increas-
1518 ing the sample size (to make it larger than the number of markers, which we are not far off
1519 in case of large GWAS analyses). Under those conditions by modifying our aim to estimate
1520 genome-wide genetic effects and bidirectional causal effects at the same time, we would fulfil all
1521 assumptions and criteria for identifiability: using the notation of the Wikipedia page we have
1522 (a) X is full rank, since all SNPs are listed only once (and pre-pruned in case of perfect LD).

1523 (b) $\Gamma = \begin{pmatrix} 1 & -\alpha_{x \rightarrow y} \\ -\alpha_{y \rightarrow x} & 1 \end{pmatrix}$ is non-degenerate since both $|\alpha_{x \rightarrow y}| < 1$ and $|\alpha_{y \rightarrow x}| < 1$. (c) The
1524 errors are i.i.d. with zero mean and finite variance-covariance matrix. (d) The parameters are
1525 identifiable since they fulfil the *rank condition*, due to the spike-and-slab genetic architecture,
1526 and the direct genetic effects on the two endogenous variables (X, Y) overlap only randomly.
1527 This ensures that $(k - k_i) \geq 1$ and in our case $n_i = 1$ for $i = 1, 2$. Therefore, the $(k - k_i) \times n_i$
1528 submatrix of Π (Wikipedia notation) is simply a column vector, which clearly has rank 1 ($= n_i$)
1529 since it is non-null.

1530 Third, this model with the priors assumed in the paper do not go further beyond the INSIDE

1531 assumption for the hidden confounders. The horizontal pleiotropy $(\gamma_u q_y + \gamma_y)$ are assume to have
1532 mean 0 and are i.i.d. across SNPs, which is essentially the same assumption as INSIDE.

1533 **Response #29:** We respectfully disagree with this statement. The key aspect of the InSIDE
1534 assumption is that the direct SNP-outcome (pleiotropic) effects are independent on the direct
1535 SNP-exposure effects. This we clearly violate, since in our settings those two effects $(\gamma_u q_y + \gamma_y)$
1536 and $(\gamma_u q_x + \gamma_x)$ are correlated since their covariance is $q_y \cdot q_x \cdot h_u^2$. Of course, if U were observed
1537 and included in a multivariable MR model, the InSIDE assumption would not be violated, but
1538 this is not the case here.

1539 Finally, are e_x, e_y and e_u assumed to be independent with each other? For the assumptions after
1540 line 99, κ_x is a vector for m genes, are the elements assumed to be i.i.d. following the same normal
1541 distribution? The model written in the paper is not complete without these specifications.

1542 **Response #30:** Thank you, we agree that these assumptions were missing and have now added
1543 them to the updated Method description.

1544 So because of the problems in these structural equations, I do not think the paper can either
1545 solve the causal directionality issue nor the confounding beyond INSIDE issue.

1546 **Response #31:** We believe that this conclusion of the reviewer was based on a misunderstanding
1547 of the aims of the model and several model settings – as described above.

1548 2. I think the derivations on pages 4-7 need to be rewritten. They are really sloppy, hard to
1549 follow and contain mathematical errors. Specially, there are three places that I want to point
1550 out. First, at line 135, the authors introduced the simplification $1 - \alpha_{y \rightarrow x} \cdot \alpha_{x \rightarrow y} \approx 1$. Do the
1551 authors assume that either $\alpha_{y \rightarrow x} = 0$ or $\alpha_{x \rightarrow y} = 0$? If not, what does that approximation exactly
1552 mean? If yes, then how is the method proposed in the paper better than other bi-directional
1553 MR approaches?

1554 **Response #32:** We agree with the reviewer that the previous derivations could have been more
1555 rigorous and hence substantially rewritten to minimise any ambiguity. To address the reviewer's
1556 comment about the approximation $1 - \alpha_{y \rightarrow x} \cdot \alpha_{x \rightarrow y} \approx 1$, we have now removed this approximation
1557 and derived the exact formulae.

1558 Second, line 142, I find the definition of π_x, π_y and π_u confusing. I think there is a typo in line
1559 142 where $\rho'_k \odot \zeta$ is not well-defined. It should be either $\rho_k \odot \zeta$, which is an m dimensional
1560 vector, or $\rho'_k \zeta$ which is a scalar. Based on the assumptions after line 99, as $\zeta_x, \zeta_y, \zeta_u$ are random,
1561 the π_x, π_y, π_u are also random as they are functions of $\zeta_x, \zeta_y, \zeta_u$, then how can they become
1562 parameters in the likelihood (after line 169)?

1563 **Response #33:** Thank you for pointing out this typo, it should have been $\rho'_k \cdot \zeta$. In the old
1564 version, we proposed to fit a (spike-and-slab) two component Gaussian mixture (both with zero
1565 and one with zero variance) to each $\rho'_k \cdot \zeta_x, \rho'_k \cdot \zeta_y, \rho'_k \cdot \zeta_u$ and denoted the mixing proportions of the
1566 component with non-zero variance with π_x, π_y, π_u , respectively. While the former definition of
1567 π s was completely legit, for a fixed k , $\rho'_k \cdot \zeta_x$ is a univariate distribution to which parameters can
1568 be fitted. Triggered by the reviewer's pertinent criticism, we do not apply such approximation
1569 to this distribution any longer, but we rather model it without any approximation, which turns
1570 out to be a binomially-weighted sum of modified Bessel functions of the second kind. Please
1571 see the description of the overall changes related to this improvement at the beginning of this
1572 Response Letter and the full details in the updated Methods section.

1573 Third, as one can already seen in the second issue, π_x, π_y, π_u are random variables. It is ap-
1574 parent that the distributions derived after line 151 are incorrect. Can the authors provide
1575 more mathematical details on how they find these to be two-component Gaussian mixtures?

1576 They are summations of Gaussian mixtures, so I don't think these results are mathematically
1577 correct.

1578 Response #34: These π s are now defined more intuitively and they are simply model parameters.
1579 The reviewer correctly points out that it is not ideal to model the sum of Gaussian mixtures as a
1580 Gaussian mixture. In the updated Methods section, we have now provide a detailed explanation
1581 how they are modelled.

Reviewers' Comments:

Reviewer #1:

Remarks to the Author:

The authors have significantly improved the method and provided reasonable justification in response to the comments made before, in particular about the model identifiability. However, there are still a number of places where the presentation is not clear.

(1) Line 363: the text says the sample size is lowered to 10,000. However, in the figure, it should be 50,000. Also, the results here (Panel b, at $N = 50K$) are not referenced in the text. Also in the same paragraph, the authors referred to Figures S5. However, that figure was obtained under a different setting (exposure and outcome sample sizes are different). This is not made clear in the text.

(2) Comparison with CAUSE in simulations: there are a number of discrepancies in this part. Simulations were done under $N = 50K$ and $500K$, and the results were reported in Fig. S14 and S15, respectively. However, the text discussed and referred to only one figure. The results of reverse simulation were reported in Fig. S16 and S17, again, the text and figure do not match.

(3) About the different results of LHC-MR vs. IVW: there are 17 pairs that were significant under IVW, but not LHC-MR. The authors suggested that these pairs are likely to be false positives. However, it is hard to verify these results. The 17 pairs are not easy to find. In Line 520: Fig. S21 cannot be found.

(4) Line 523, Table S7: this table is not informative. It only shows the statistics of the pairwise comparison of methods. It doesn't list the exact pairs here, and the relevant results.

(5) Comparison with CAUSE, Line 561: overall I feel that the results are not well summarized in the text. The main result is shown in Table 1: when both CAUSE and LHC report significant results, the effects are correlated. But a more interesting result would be whether the two methods reported similar causal pairs. No information is given in the main text. In the Supple. table, one can actually see that the results from the two methods agree reasonably well, e.g. among top 10-20 pairs by LHC, most are also found by CAUSE. I feel that it would be helpful to give a better summary here.

Reviewer #2:

Remarks to the Author:

The authors have addressed my concerns and the paper is now considerably improved. The method itself is improved and more comprehensive and realistic simulations have been conducted. The authors and/or the editor may want to polish the writing in places, e.g. in the second sentence "based on observational data due to its abundance over the years". The "over the years" here is a bit vague and unnecessary. Polishing is not necessary but may improve readability and the reception of the paper.

Reviewer #3:

Remarks to the Author:

With the revised paper and the authors' response, I think the paper is much improved and I now can appreciate the ideas and model in the paper. I think the authors have addressed most of my previous two concerns. The only remaining concern I have is that I'm still not quite convinced that the equations on page 3 are structural equations and the $\alpha_{y \rightarrow x}$ and $\alpha_{x \rightarrow y}$ represent causal effects. Typically, structural equations do not contain loops and it's hard to imagine how the data can be generated. I would suggest the authors add some references to support their way of defining the structural equations and the definition of the causal effects.

Also there is a typo in the equations on pages 4-5. All the $\alpha_{y \rightarrow x} \alpha_{y \rightarrow x}$ or

$\alpha_{x \to y} \alpha_{x \to y}$ should be $\alpha_{y \to x} \alpha_{x \to y}$

Finally, I think the authors may reorganize the paper to put some mathematical details in the supplement instead of the Methods section and write a longer method overview section in Section 3.1 to summary their idea and model setup.

Reviewer Comments - part 2

Overall responses: We sincerely thank the reviewers for going through our heavily updated manuscript and taking the time to give us feedback. We are glad that the changes made in the method and the subsequent analyses were well received, and we hope that the additional changes done in response to their latest comments are satisfactory. For better readability, we will be referring to the different versions of the manuscript in the responses below as follows:

- V1 - Originally submitted manuscript, fall 2020.
- V2 - Re-submitted manuscript responding to first round of reviews, summer 2021.
- V3 - Updated manuscript following second round of reviews and editorial requests, September 2021.

Reviewer #1 (Remarks to the Author):

The authors have significantly improved the method and provided reasonable justification in response to the comments made before, in particular about the model identifiability. However, there are still a number of places where the presentation is not clear.

We thank the reviewer for their kind comment, and again for their original comment that prompted several of the updates that make the updated method more robust. Unfortunately, the comments made by the reviewer below are often referring to the older version of the manuscript (V1), possibly due to a confusion between the two versions. Most of the comments have already been addressed in the new manuscript (V2) and we will highlight these updates below.

(1) Line 363: the text says the sample size is lowered to 10,000. However, in the figure, it should be 50,000. Also, the results here (Panel b, at $N = 50K$) are not referenced in the text. Also in the same paragraph, the authors referred to Figures S5. However, that figure was obtained under a different setting (exposure and outcome sample sizes are different). This is not made clear in the text.

Line 363 in V1 indeed does have a discrepancy between the sample sizes reported in the figures and the text. In V2 however, we limited the sample size (N) to 50,000 and 500,000, and repeated each scenario with these two different sample sizes, this is mentioned in lines 99-101 of V3 (lines 402-404 in V2 before the editorial updates). Indeed Figure S5 still refers to testing a scenario where the two samples sizes for the exposure and the outcome differ in size, and we hope that the newer manuscript is clear in indicating these two differing sample sizes in lines 118-123 of V3 (421-426 in V2).

(2) Comparison with CAUSE in simulations: there are a number of discrepancies in this part. Simulations were done under $N = 50K$ and $500K$, and the results were reported in Fig. S14 and S15, respectively. However, the text discussed and referred to only one figure. The results of reverse simulation were reported in Fig. S16 and S17, again, the text and figure do not match.

We refer to figures S14 and S15 in lines 213 and 218 respectively (514 and 520 in the pre-editorial updates). These two figures represent the results of running LHC-MR simulated data under the CAUSE model, with both sample sizes. Each of these figures illustrates three scenarios, the first being LHC-MR data generated with no confounder, then with no causal effect, and finally with both a confounder and a causal effect. Figures S16 and S17, which are mentioned in lines 221 and 237 (523 and 538 in V2) respectively of V3, represent the inverse scenario where data is generated using the CAUSE framework and ran under the LHC-MR model. The same three scenarios are applied for the differing sample sizes.

(3) About the different results of LHC-MR vs. IVW: there are 17 pairs that were significant under IVW, but not LHC-MR. The authors suggested that these pairs are likely to be false positives. However, it is hard to verify these results. The 17 pairs are not easy to find. In Line 520: Fig. S21 cannot be found.

Although this observation is referring to previous results in the V1 of the manuscript, we agree with the reviewer about the interpretation of these pairs. In the revised analysis (V2/V3), there are now seven pairs that are significant for IVW but not for LHC-MR, as opposed to the previous 17. Of these seven pairs, four of them are only picked up by IVW and not the other methods, thus making us inclined to believe that they may be false positives. This comparison can be more easily seen in Table S7 where a visual comparison is available for all MR methods as well as CAUSE. The reviewer's comment about not finding figure S21 in line 520 is referring to V1, which now does not match the number of figures in the new supplementary materials, however we have already amended this issue in V2/V3.

(4) Line 523, Table S7: this table is not informative. It only shows the statistics of the pairwise comparison of methods. It doesn't list the exact pairs here, and the relevant results.

Although this comment also references V1 and the previous Table S7, we agree with the reviewer that the comparison done was uninformative, and thus have dropped the table in the new version. In our new reporting, we focus on the comparison between the significance status of each method for the different pairs, as well as the direction of the causal effect if it is significant. We try to understand and find supporting evidence for the concordance in significance and direction between methods instead of focusing on the statistically significant difference between their reported causal effects.

(5) Comparison with CAUSE, Line 561: overall I feel that the results are not well summarized in the text. The main result is shown in Table 1: when both CAUSE and LHC report significant results, the effects are correlated. But a more interesting result would be whether the two methods reported similar causal pairs. No information is given in the main text. In the Supple. table, one can actually see that the results from the two methods agree reasonably well, e.g. among top 10-20 pairs by LHC, most are also found by CAUSE. I feel that it would be helpful to give a better summary here.

Paragraph starting at line 348 (649 in V2) in V3 corresponds to the updated real-data results comparing LHC-MR causal estimates with those from CAUSE (paragraph starting at 561 in V1). These results have been updated and reworded in the newer manuscript, and are now accompanied by Figure 19 in the supplementary material where the correlation between the results of the two methods is more clearly seen. Moreover, Table S6 in the supplementary materials also shows a visual comparison between all the MR methods, including CAUSE and LHC-MR, on whether they are significant or not, and if they are, then in which direction. We hope that these supplementary materials convey the comparison between the two methods sufficiently.

Reviewer #2 (Remarks to the Author):

The authors have addressed my concerns and the paper is now considerably improved. The method itself is improved and more comprehensive and realistic simulations have been conducted. The authors and/or the editor may want to polish the writing in places, e.g. in the second sentence “based on observational data due to its abundance over the years”. The “over the years” here is a bit vague and unnecessary. Polishing is not necessary but may improve readability and the reception of the paper.

We thank the reviewer for their kind comments. We do believe the readability of the paper can be improved and have taken measures to first re-structure the paper and shift the methodological details to the supplementary material, and have gone over the introduction again to reduce ambiguity.

Reviewer #3 (Remarks to the Author):

With the revised paper and the authors’ response, I think the paper is much improved and I now can appreciate the ideas and model in the paper. I think the authors have addressed most of my previous two concerns. The only remaining concern I have is that I’m still not quite convinced that the equations on page 3 are structural equations and the $\alpha_{y \rightarrow x}$ and $\alpha_{x \rightarrow y}$ represent causal effects. Typically, structural equations do not contain loops and it’s hard to imagine how the data can be generated. I would suggest the authors add some references to support their way of defining the structural equations and the definition of the causal effects.

There is indeed relatively limited literature on non-recursive SEM, but they do exist [see Finch, W. and French, B. (2015)¹, and references within]. In our opinion the problem of loops in the SEM setting can be approached from two angles. First, the classical (recursive) SEM framework can accommodate such situations by allowing multiple instances of a given trait, representing their realisations at different time points². This avoids loops, but adds more variables, and the observed variables are the ones that represent the variables as time tends to infinity (equilibrium state), while data from previous time points are latent. A second approach, which we took, is to convert the originally “looping” equations (Eqs. 1-3) to “non-looping” equations (Eqs. 10-11). As the reviewer can appreciate that these latter equations also directly allow data generation. Interestingly, the latter equations can be obtained also by the first approach, by computing the limit as time goes to infinity (it is a generalisation of the infinite sum of a geometric series of type $\sum_{i=0}^{\infty} q^i = (1 - q)^{-1}$). Finally, we agree with the reviewer that such looping equations render parameter interpretation inconvenient and were glad to see that for real data applications such loops were rare.

Also there is a typo in the equations on pages 4-5. All the $\alpha_{y \rightarrow x} \alpha_{y \rightarrow x}$ or $\alpha_{x \rightarrow y} \alpha_{x \rightarrow y}$ should be $\alpha_{y \rightarrow x} \alpha_{x \rightarrow y}$

This is a very helpful catch from the reviewer, thank you! We have now updated the equations to correct for these typos. The change is not highlighted, as it is cumbersome to highlight mathematical equations, but the equations between lines 552 and 595 in V3 (97 and 143 in the V2) have been amended.

Finally, I think the authors may reorganize the paper to put some mathematical details in the

¹W. Holmes Finch & Brian F. French (2015) Modeling of Nonrecursive Structural Equation Models With Categorical Indicators, *Structural Equation Modeling: A Multidisciplinary Journal*, 22:3, 416-428, DOI: 10.1080/10705511.2014.937380

²Wong C-S, Law KS. Testing Reciprocal Relations by Nonrecursive Structural equation Models Using Cross-Sectional Data. *Organizational Research Methods*. 1999;2(1):69-87. doi:10.1177/109442819921005

supplement instead of the Methods section and write a longer method overview section in Section 3.1 to summary their idea and model setup.

We agree with the reviewer that the methods section can be re-arranged to include some more details in the supplementary materials while keeping the main text light enough to aid with readability. We have now merged section 2.3-“Characteristic functions of $z_k^{(u)}$, $z_k^{(x)}$, $z_k^{(y)}$ and (η_k^x, η_k^y) ” and 2.4-“From characteristic function to probability density function” into a single section called “Derivation of the likelihood function” (and shortened it substantially by moving many mathematical derivations to Supplementary Methods 1.1-1.2). Furthermore, we have also summarised section 2.6-“Decomposition of genetic correlation”, and have moved its full version to Supplementary Methods 1.5 along with section 2.7-“Computation of the LD scores”.